# Chronostratigraphy of Larsen blue ice area in Northern Victoria Land, East Antarctica, and its implications for paleoclimate

Giyoon Lee[1], Jinho Ahn[1], Hyeontae Ju[2], Florian Ritterbusch[3], Ikumi Oyabu[4], Christo Buizert[7], Songyi Kim[2], Jangil Moon[2], Sambit Ghosh[1], Kenji Kawamura[4,5,6], Zheng-Tian Lu[3], Sangbum Hong[2], Chang Hee Han[2], Soon Do Hur[2], Wei Jiang[3], and Guo-Min Yang[3]

[1]School of Earth and Environmental Sciences, Seoul National University, Seoul, South Korea
[2]Korea Polar Research Institute, Incheon, South Korea
[3]Hefei National Laboratory for Physical Sciences at the Microscale, CAS Center for Excellence in Quantum Information and Quantum Physics, University of Science and Technology of China, Hefei, China.
[4]National Institute of Polar Research, Tachikawa, Japan
[5]Department of Polar Science, School of Multidisciplinary Sciences, The Graduate University for Advanced Studies, SOKENDAI, Tachikawa, Japan
[6]Japan Agency for Marine-Earth Science and Technology (JAMSTEC), Yokosuka, Japan
[7]College of Earth, Ocean and Atmospheric Sciences, Oregon State University (OSU), Corvallis, OR, USA

*Correspondence to*: Jinho Ahn (jinhoahn@gmail.com)

**Abstract.** Blue ice areas (BIAs) allow for the collection of large-sized old ice samples in a cost-effective way because deep ice outcrops and make old ice samples available close to the surface. However, chronostratigraphic studies on blue ice areas are complicated owing to fold and fault structures. Here, we report a surface transect of ice with an undisturbed horizontal stratigraphy from the Larsen BIA, Antarctica, making the area valuable for paleoclimate studies. Ice layers defined by dust bands and ground penetration radar (GPR) surveys indicate a monotonic increase in age along the ice flow direction on the downstream side, while the upstream ice exhibits a potential repetition of ages on scales of tens of meters, as shown in complicated fold structure. Stable water isotopes ($\delta^{18}O_{ice}$ and $\delta^2H_{ice}$) and components of the occluded air (i.e., $CO_2$, $N_2O$, $CH_4$, $\delta^{15}N-N_2$, $\delta^{18}O_{atm}$ (= $\delta^{18}O-O_2$), $\delta O_2/N_2$, $\delta Ar/N_2$, $^{81}Kr$ and $^{85}Kr$) were analyzed for surface ice and shallow ice core samples. Correlating $\delta^{18}O_{ice}$, $\delta^{18}O_{atm}$, and $CH_4$ records of Larsen ice with existing ice core records indicate that the gas age at the shallow coring sites ranges between 9.2–23.4 ka BP, while the ice age for the entire surface sampling sites between 5.6–24.7 ka BP. Absolute radiometric $^{81}Kr$ dating for the two cores confirms ages within acceptable levels of analytical uncertainty. A tentative climate reconstruction suggests a large deglacial warming of 15 ± 5 °C that we attribute to the retreat of the Antarctic ice sheet from the adjacent Ross Sea embayment, and an increase in snow accumulation by a factor of 1.7–4.6 due to the increased penetration of snow-bearing storms. Climatic interpretation is complicated by the need for upstream flow corrections, evidence for strong surface sublimation during the last ice age, and potential errors in the estimated gas age-ice age difference. Our study demonstrates that BIAs in Northern Victoria Land may help to obtain high-quality records for paleoclimate and atmospheric greenhouse gas compositions through the last deglaciation.

## 1 Introduction

Ice cores serve as very useful archives for paleoclimate records, such as ancient atmospheric greenhouse gas composition, surface temperature, and aerosols. For instance, ice core records reveal the relationship between greenhouse gas concentration and Earth's climate (Petit et al., 1999; EPICA Community Members, 2004; Siegenthaler et al., 2005; Lüthi et al., 2008; Bereiter et al., 2015), abrupt climate changes during the last glacial period (Dansgaard et al., 1989; Steffensen et al., 2008), and bipolar seesaw climate links on millennium timescales (Blunier and Brook, 2001; Landais et al., 2015). Thus far, continuous climate records from ice cores cover the last 800 ka and may reach more than 1 Ma in the near future by new deep drilling projects in the Antarctica (Fischer et al., 2013). However, the use of ice cores obtained from conventional deep ice core drilling projects remains limited as many analyses require large amounts of ice, such as trace element isotope and trace gas analyses. In addition, deep drilling projects incur high economic and labour costs. In contrast, coring in blue ice areas (BIAs) has emerged as an alternative to obtain large ice samples in a cost-effective manner (Folco et al., 2006; Petrenko et al., 2006; Schaefer et al., 2006; Sinisalo et al., 2007; Korotkikh et al., 2011; Turney et al., 2013; Bauska et al., 2016; Aarons et al., 2017; Baggenstos et al., 2017; Zekollari et al., 2019; Yan et al., 2019; Fogwill et al., 2020).

Once ice is deposited, it flows to the margin of the ice sheet, where it is exposed on the surface since the basal topographic obstacles cause deep glacial flow upward. Moreover, surface snow is ablated by katabatic winds and/or sublimation (Bintanja, 1999; Sinisalo and Moore, 2010). Because ice layers of the same age (isochrones) are extended on the surface, we can obtain large amounts of old ice for a specific age at the surface and/or relatively shallow depths, allowing researchers to study paleoclimate which is typically prohibited by limited sample sizes (Schaefer et al., 2006; Bauska et al., 2016; Fogwill et al., 2020).

In the early stages of BIA research, meteorites were the focus of investigation because ice flow and ablation cause meteorites to accumulate on the surface of the BIAs (Whillans and Cassidy, 1983; Cassidy et al., 1992, Harvey, 2003). Recently, studies on BIAs have drawn attention to the possibility of identifying ice older than 800 ka BP (to date, the longest continuous ice core record covers the last 800 ka BP) because glacial–interglacial cycles changed from 40 to 100 kyr during the 0.8–1.0 Ma period, which is called the mid-Pleistocene Transition (MPT). It has been reported that ice at Allan Hills BIA has ages of 90–250 ka BP on the surface (Spaulding et al., 2013), reaching ~2.7 Ma BP near the bedrock at depths of about 150–200 m (Yan et al., 2019). However, the use of ice samples taken from BIAs (hereafter referred to as blue ice) has several drawbacks. In most cases, the stratigraphy of the BIAs is complicated, as shown by the fold and fault structures on the surface (Folco et al., 2006; Petrenko et al., 2006; Curzio et al., 2008; Schaefer et al., 2009; Baggenstos et al., 2017) and stratigraphy is discontinuous at deep depths near the bedrock where ice ages are similar to or older than the MPT (Higgins et al., 2015; Yan et al., 2019).

BIAs cover 1.67 % of the Antarctic continent and are concentrated in Victoria Land, the Transantarctic Mountains, Dronning Maud Land and the Lambert Glacier basin (Hui et al., 2014) (Fig. 1). Several chronological studies of BIAs have been conducted in Southern Victoria Land: Taylor Glacier (Aciego et al., 2007; Buizert et al., 2014; Baggenstos et al., 2017; Baggenstos et al., 2018; Menking et al., 2019), Allan Hills (Spaulding et al., 2013; Higgins et al., 2015; Yan et al., 2019; Yan

et al., 2021), and Mullins Glacier (Yau et al., 2015); Northern Victoria Land: Frontier Glacier (Folco et al., 2006; Curzio et al., 2008; Welten et al., 2008). Studies on the Taylor Glacier and Allan Hills BIA have constrained age along several transects and cores. However, chronologies of the other BIAs in Victoria Land remain insufficient for high-resolution paleoclimate studies. In addition, important paleoclimate proxies such as stable water isotopes, greenhouse gas , and isotopic ratio of the oxygen gas
have not been previously addressed for the BIAs in Northern Victoria Land.

Generally, because of the complicated stratigraphy, blue ice samples are dated using multiple methods (Sinisalo and Moore, 2010). In previous studies, ice flow modelling, correlation analysis with stable isotopes of ice, and radiometric analysis of meteorites and tephra in ice have been used, but the age constraints have not been sufficiently precise to study paleoclimate (Azuma et al., 1985; Nakawo et al., 1988; Reeh et al., 2002; Folco et al., 2006; Aciego et al., 2007; Curzio et al., 2008; Dunbar
et al., 2008). One effective method for dating the gas age is to correlate globally well-mixed atmospheric gas records (e.g., $CH_4$, $CO_2$, $\delta^{18}O_{atm}$) with existing well-dated ice core records (Spaulding et al., 2013; Baggenstos et al., 2017; Menking et al., 2019; Yan et al., 2021). Other methods include the use of stable Ar isotopes (Higgins et al., 2015; Yau et al., 2015; Yan et al., 2019) or radioactive $^{81}Kr$ (Loosli&Oeschger 1969; Buizert et al., 2014; Tian et al., 2019; Crotti et al., 2021), both of which provide independent and absolute age constraints. However, the use of Ar and Kr isotopes have certain limitation for accurate
age constraint. The age uncertainty of Ar dating is ±180 ka or 11 % of the age; the uncertainty is originated from the regression line Bender et al. (2008) used. In addition, the ages can be corrupted by the injection of radiogenic $^{40}Ar$ from the continental crust (Bender et al., 2010). The age uncertainty of $^{81}Kr$ dating ranges between 5–20 % of the age depending on the sample age and sample size (Jiang et al., 2020). It also has a systematic age uncertainty of ~5 % due to the uncertainty in the $^{81}Kr$ half-life. For dating ice ages, glaciochemical records (e.g., nss-$Ca^{2+}$, $\delta^{18}O_{ice}$, $\delta^2H_{ice}$) can be used for correlation with existing well-dated
ice core records (Baggenstos et al., 2018; Menking et al., 2019). Notably, the ice is older than the gas at the corresponding depth because the gas is isolated and stops mixing with the atmospheric air when the firn completely transforms into ice (Schwander and Stauffer, 1984).

$CH_4$ and $O_2$ are well mixed in the atmosphere (Blunier et al., 2007), and conducting correlations of both $CH_4$ concentration and $\delta^{18}O$ of $O_2$ ($\delta^{18}O_{atm}$) is a well-known strategy for establishing chronologies of blue ice (Petrenko et al., 2006; Baggenstos
et al., 2017; Yan et al., 2021). Variations in continental ice mass are known to be the main factor that controls $\delta^{18}O_{atm}$ during glacial−interglacial cycles (Bender et al., 1985; Sowers et al., 1993). In addition, variations in the Inter Tropical Convergence Zone (ITCZ) were also considered to control $\delta^{18}O_{atm}$ on millennial and orbital timescales (Severinghaus et al., 2009; Landais et al., 2010; Seltzer et al., 2017; Extier et al., 2018b). Because of the long lifetime of oxygen gas in the atmosphere (~1 ka), $\delta^{18}O_{atm}$ varies more gradually, limiting synchronization to millennial timescales. In contrast, atmospheric $CH_4$ concentration
changes rapidly because of the short lifetime of ~12 years, allowing precise dating via stratigraphic matching. Oeschger (1987) suggested the potential of $^{81}Kr$ measurements in ice core dating. However, at that time, $10^5$–$10^6$ kg of ice was required. Owing to the development of Atom Trace Trap Analysis (ATTA), the required ice has continued to decrease (Lu et al., 2014; Tian et al., 2019; Jiang et al., 2020; Crotti et al., 2021). Buizert et al. (2014) for the first time, showed that $^{81}Kr$ age dating is feasible for blue ice in Taylor Glacier, Antarctica.

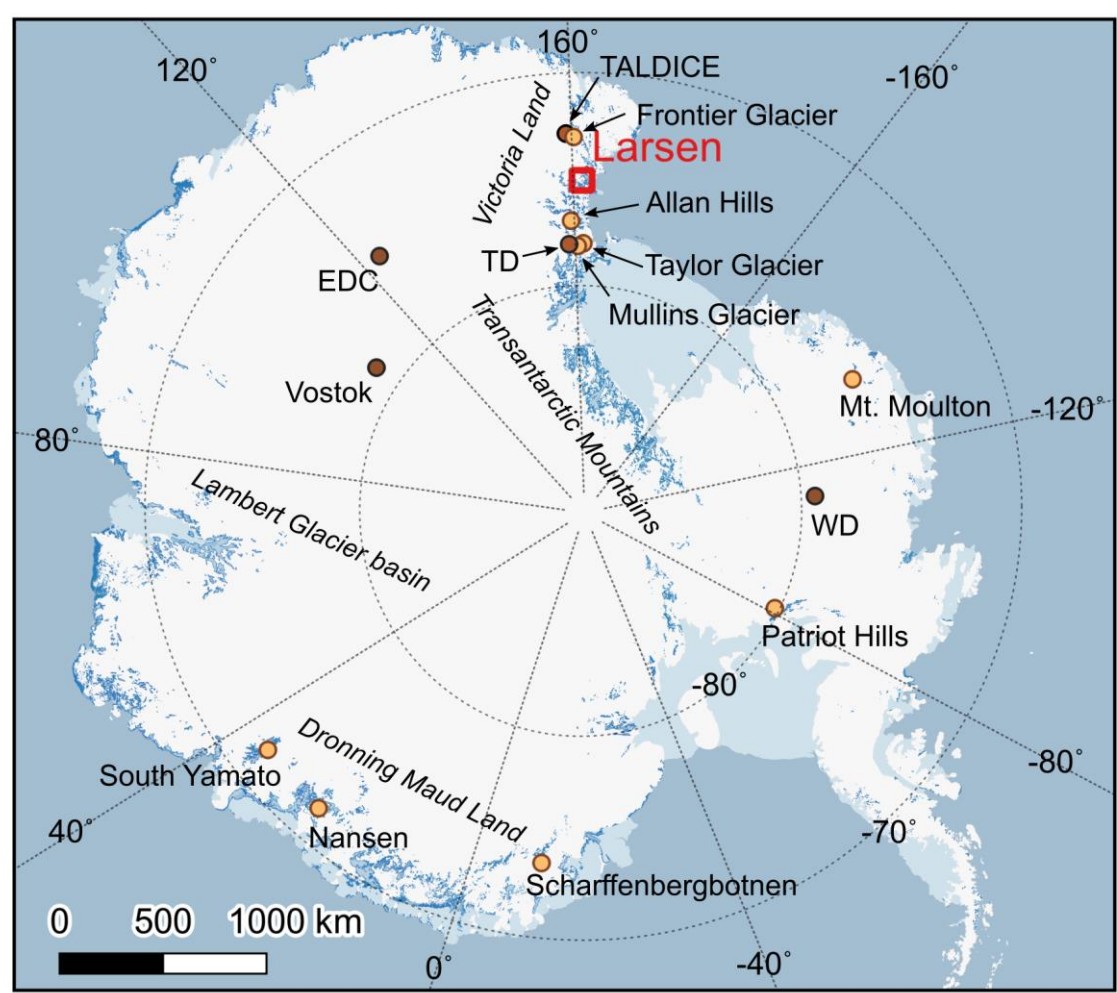


**Figure 1. Distribution map of BIAs in Antarctica.** BIAs are noted in dark blue (Hui et al., 2014). Orange dots represent the BIAs where the chronology has been studied. Deep ice core locations with brown dots are labeled as follows: Talos Dome ice core (TALDICE), Taylor Dome (TD), EPICA Dome C (EDC), WAIS Divide (WD) and Vostok. To keep a consistent orientation with the GPR profile in Fig. 3, we flipped the classical map of Antarctica (East Antarctica to the left-hand side). The Antarctic map was obtained from the QGIS Quantarctica

package.

Our study focuses on the chronostratigraphy of ice in Larsen BIA, Antarctica, which may facilitate future research in this region. We describe the ice flow and structure of an ice body using dust bands and ground penetration radar (GPR) surveys, and then assessed the alterations of the measured stable water isotopes, greenhouse gases ($CH_4$, $CO_2$), and gas isotopes ($\delta^{15}N$-$N_2$, $\delta^{18}O_{atm}$). To constrain the unknown gas and ice ages, $\delta^{18}O_{atm}$, $CH_4$, and $\delta^{18}O_{ice}$ were correlated with existing ice core

records. We also independently confirmed the ages using the radiometric $^{81}Kr$ dating method. Finally, using the $\delta^{15}N$-$N_2$, and the $\Delta$age (ice age-gas age difference) results, we present the record of surface temperature and accumulation rate. In contrast to the previous studies, which used the Herron-Longway model (Herron and Langway, 1980; Baggenstos et al., 2018; Menking

et al., 2019; Yan et al., 2021), we applied a recently developed analytical framework to estimate past surface temperatures and accumulation rates (Buizert, 2021).

## 2 Study area and methods

### 2.1 Larsen blue ice area (BIA)

We sampled ice from Larsen BIA, an outlet glacier in the Northern Victoria Land, East Antarctica, during the austral summer of 2018/2019. The Larsen BIA is located approximately 85 km southwest of the Korean Jang Bo Go station (Fig. 2a). The mean annual temperature is $-24.4 \pm 11.7$ °C (data and information were obtained from the Meteo-Climatological Observatory at MZS and Victoria Land of PNRA-www.climantartide.it), cold enough to prevent ice melting in the summer. There were ~20 cm wide dust bands with gentle folding structures in the mid- to downstream part, while we observed severely folded dust bands (e.g. S- and Z-folds) in the upstream part (Fig. 2b). Dust bands are frequently observed in BIAs in Victoria Land and can be used as isochrons (Sinisalo and Moore, 2010). To obtain ice samples with simple stratigraphy, we avoided ice coring in areas with complex fold structures and sampled ice in the direction of ice flow identified by the Antarctic ice velocity map in the QGIS Quantarctica package (Rignot et al., 2011; Mouginot et al., 2012; Matsuoka et al., 2018; Matsuoka et al., 2021). However, the stratigraphy of the upstream ice, where we collected surface ice may be inverted and repeated on a scale of tens of meters. Shallow ice cores were drilled along a 1 km long transect at intervals of 20–30 m (Fig. 2b). Most ice cores had lengths of approximately 2 m, but ice core #23 (74.9319° S, 161.6018° E) was ~10.4 m (Fig. 2c). We also collected near-surface ice samples (~500 g) along a 1.3 km transect at 20 m intervals to measure stable water isotopes of the ice ($\delta^{18}O_{ice}$, $\delta^2H_{ice}$) at depths of ~5–10 cm (hereafter referred to as the surface ice sample). The ice core TF (74.93042° S, 161.56975° E) with a length of ~12 m on the upstream side was sampled in 2016 for a preliminary study, as described by Jang et al. (2017). An imaginary line parallel to the ice flow direction was used to define the horizontal distance, while a perpendicular line from each sampling location to the line parallel to the ice flow direction was projected to identify the intersection point. Each intersection point was then used to measure the horizontal distance from the most upstream sampling site (Fig. S1).

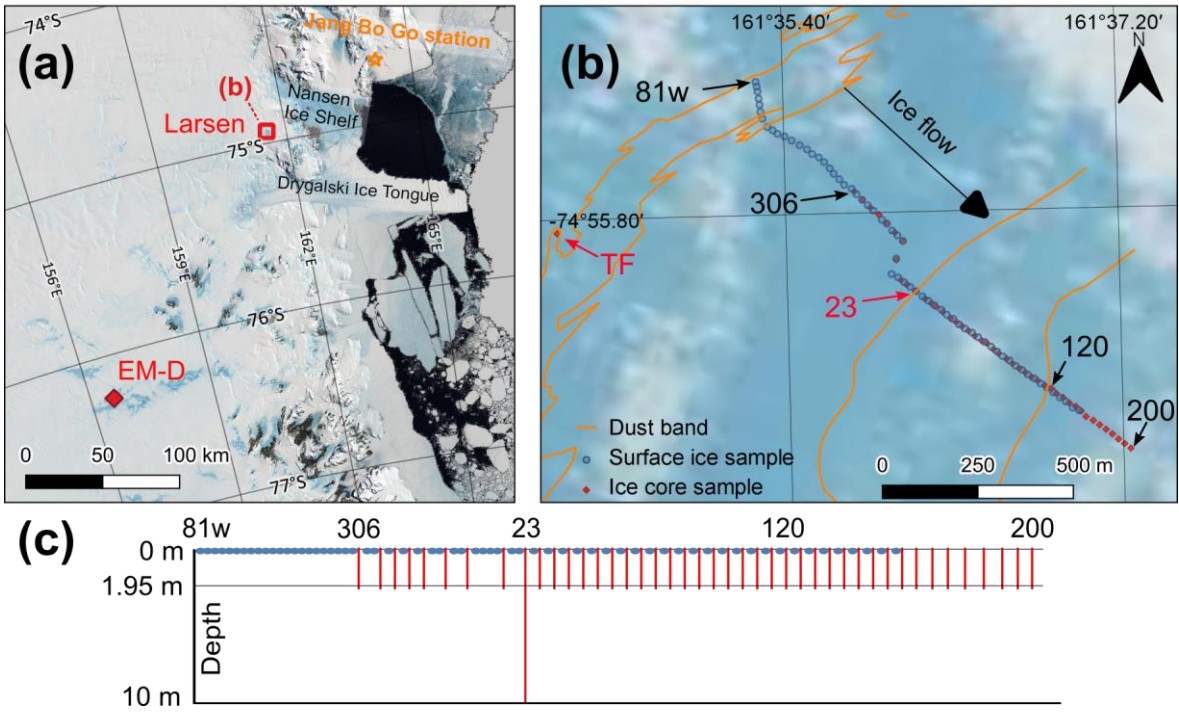

**Figure 2. Location of the Larsen BIA and sample collection.** (a) Location of Larsen BIA, EM-D core, and Jang Bo Go station. (b) Magnified map of Larsen BIA including sample locations. Orange lines represent dust bands observed in the blue ice field. The line marks are derived from what appeared on Google Earth. Blue dots represent locations of surface ice samples. Red diamonds are locations of shallow ice cores. Six representative names are shown, red letters for ~10 m long cores (TF and #23), black letters for 2 m long cores (#306, #120, and #200) and surface ice (81w). Z- and S-folds are recognized by the dust bands at upstream ices. The total transect of the ice sample is approximately 1.4 km. (c) Schematized cross-section of the transect. Satellite photo of Antarctica is from the QGIS Quantarctica package (Bindschadler et al., 2008).

## 2.2 Ground penetrating radar (GPR) survey

Approximately 17 km of GPR data was collected for two days in January 2019 (Fig. S1). A MALÅ ProEx impulse radar system with a 50 MHz unshielded antenna for a larger penetration depth was used for data acquisition. Records were obtained at a sampling frequency of 559.5 MHz, time interval 0.1 s and stacked 4 times. The survey was conducted at a speed of 2.5–3 km h$^{-1}$ to minimize noise caused by frictional vibrations between the antenna and the surface of the glacier. The position of traces was recorded using a single-frequency code-phase GPS with an accuracy of < 3 m. Data processing was performed in ReflexW v.9.5 in the order of dewow, DC filter, band-pass filter, time-zero drift, energy decay correction, background removal, static correction, migration, and stack. As the constant radar signal velocity "v" in the glacier was 0.17 m ns$^{-1}$ (Borgorodsky, 1985; Reynolds, 1985), and the frequency "f" of the radar system was 50 MHz, the GPR wavelength "λ" was about 3.4 m (λ = v f$^{-1}$).

## 2.3 Stable water isotope measurement

Stable water isotopes ($\delta^{18}O_{ice}$, $\delta^2H_{ice}$) were measured simultaneously at the Korea Polar Research Institute (KOPRI) using

the Cavity Ring-Down Spectroscopy (CRDS) method using a Picarro L2130-i with a vaporizer (A0211). Surface ice samples (~5–10 cm depth) and ice core samples (10–30 cm and 190–200 cm depths of ice cores) were used for the analyses. The average horizontal spacing of the surface ice samples and ice cores was approximately 10.3 m. Ice from the ~10 m long core #23 was used for measurements at 20 cm vertical intervals. Ice samples were kept in Whirl-Pak and melted at room temperature. The melted sample was then injected into a 2 ml vial via a disposable syringe with a 0.45 µm filter. One batch of measurements

consisted of five duplicate samples (10 samples in total) and a working standard. The first and second aliquots of the sample were measured 12 times each, while the working standard was measured 20 times. The last six measurements were used to remove memory effects from the previous samples. The precision was evaluated by measuring the working standard repeatedly (n = 67); 1 sigma (standard deviation) was 0.07 ‰ for $\delta^{18}O_{ice}$ and 0.90 ‰ for $\delta^2H_{ice}$. The working standards and samples were calibrated against the international standards for water isotopes, VSMOW2 (Vienna Standard Mean Ocean Water 2), SLAP2

(Standard Light Antarctic Precipitation 2) and GISP (Greenland Ice Sheet Precipitation).

## 2.4 Greenhouse gas (GHG) measurement

The $CH_4$ concentrations in ice cores (#306, #23, #120, and #201) were analyzed at Seoul National University (SNU) using a melt–refreeze technique with a flame ionization detector gas chromatograph (FID-GC) at 10 cm vertical intervals. The $CH_4$ measurement process is described in detail by Yang (2019). Briefly, we trimmed the outermost side and the cracks of the ice

samples by approximately 2 mm with a clean band saw to eliminate contamination by ambient air. The mass of the ice used for measurement was about 35–55 g. Then, we placed the ice inside a custom-made flask and evacuated the flask. We used a 740.6 ppb standard air sample from the National Oceanic and Atmospheric Administration (NOAA) Global Monitoring Division (GMD) on the WMO X2004A scale to establish a daily calibration line (Dlugokencky et al., 2005). The NOAA standard air was injected into four flasks containing bubble-free ice samples, which served as the control group. The $CH_4$

mixing ratio of the control group was measured, and the daily average offset to the standard value of 740.6 ppb was considered as a systematic error; thus, the daily average offset (which is in the range of 2–20 ppb) was subtracted from each measurement result. The trapped gas in the ice was liberated by immersing the flask in hot water and then refreezing the melt water. In addition, as $CH_4$ is more soluble than the major air components ($N_2$, $O_2$, and Ar), the $CH_4$ concentration of the 1st extracted air is lower than the original value. Therefore, 2nd gas extraction was conducted (refrozen meltwater was melted and refrozen once

again) to correct the solubility effect.

The same ice cores (#306, #23, #120, and #201) were used for $CO_2$ measurements. Generally, $CO_2$ should be measured using a dry extraction system rather than wet extraction because of its high solubility in water and the possibility of $CO_2$ production by carbonate–acid reactions in ice melt (Delmas et al., 1980). Therefore, for our measurement, air was collected using a needle-crusher dry extraction instrument and detected by FID-GC at SNU. The ice was pre-treated in the same manner

as that for $CH_4$, as described above, while the required amount of ice was 15–20 g. We placed the ice in the needle crush chamber, which was cooled to −35 ℃ during ice preparation. The $CO_2$ results could be affected by adsorption and desorption by the instrument and the sample tube (Ahn et al., 2009). Therefore, before crushing the ice, NOAA standard air was released through the chamber and collected into the sample tube. The $CO_2$ mixing ratio was measured, and the average offset with respect to NOAA standard air was used to subtract the value of the $CO_2$ result from each ice sample measurement. The $CO_2$

mixing ratio of the standard air we used for making a daily calibration line was 285.66 ppm or 293.32 ppm, which is from NOAA GMD in WMO X2019 calibration scale (Hall et al., 2021). Trapped air in the ice was collected in a sample tube cooled by a helium closed cycle refrigerator (He-CCR) after crossing the water trap (approximately −80 ℃). The entire process is described in detail by Shin (2014).

    We also used ice samples from 35 cores at depths of 190–200 cm (hereafter referred to as horizontal measurement) and ice

core #23 to measure $CH_4$, $CO_2$, and $N_2O$ concentrations, along with other gas isotopes (see below), by a wet extraction method at the National Institute of Polar Research (NIPR) in Japan. For $CO_2$ and $CH_4$, FID-GC was used, and for $N_2O$, an electron capture detector (ECD) GC was used. To determine the concentrations of $CH_4$ and $CO_2$, a calibration line was established using TU-2008 (Tohoku University) scale standard air; for $N_2O$, the TU-2006 scale standard air was used. Details on the standard air used in this study (named STD 1, 3, 5) are described by Oyabu et al. (2020). Three standard air and a quadratic

calibration line were used to determine the GHG concentration in the NIPR, while single standard air and a linear calibration line were used to determine the GHG concentration in SNU. Different calibration scales of standard air (TU and NOAA/WMO) and different calibration methods may contribute to making an inter-laboratory offset of GHG concentrations. $CO_2$ was measured again at SNU using the dry extraction method at 190–195 cm depth for the horizontal measurement (Table S4). The ice preparation process is briefly explained in Sect. 2.5.

Here, we also report $CH_4$ and $CO_2$ measurement results of EM-D core for comparison (76.253° S, 156.562° E) (Fig. 2), which was sampled on 13 December, 2016 at Elephant Moraine Texas Bowl for preliminary study (Jang et al., 2017).

## 2.5 Analyses of $\delta^{15}$N-$N_2$, $\delta^{18}O_{atm}$, $\delta O_2/N_2$, and $\delta Ar/N_2$

    $O_2$, $N_2$, and Ar isotopes were measured at the NIPR using a dual-inlet mass spectrometer (Thermo Fisher Scientific Delta V). The isotope results were evaluated with respect to the modern atmosphere. As mentioned previously, we measured 35 ice

cores at depths of 190–200 cm to evaluate how ancient air compositions change horizontally and also measured gas isotopes using ice core #23 at several depths with a range of 0–10 m to compare with the results from the neighbouring cores at depths of 190–200 cm. Because gas loss fractionation by molecular diffusion during storage could have affected the isotope results (Ikeda-Fukazawa et al., 2005; Oyabu et al., 2021), we trimmed the surface and the cracks of the ice approximately 3–5 mm and then shaved away some blurry ice surface using a ceramic knife. The weight of the ice obtained was 55–80 g after trimming

the surfaces. The ice inside the vessel was evacuated for ~120 min. To extract the gas, each vessel was gradually immersed in a hot water container. Simultaneously, the released air was collected into a sample tube, which was cooled by a He-CCR. After

homogenizing the sample tube overnight, the gas was split into two aliquots, one for isotope analysis (1.5 ml) and the other for greenhouse gas measurement (5 ml). The measurement process is described in more detail by Oyabu et al. (2020).

Obtaining a true $\delta^{18}O$ value in the past atmosphere from ice cores requires gravitational, thermal, and gas loss fractionation corrections. The gravitational factor is proportional to the difference in the mass number between isotopes (Craig et al., 1988; Severinghaus et al.,1998). Therefore, $\delta^{18}O$ ($^{18}O/^{16}O$) was affected twice as much as that of $\delta^{15}N$ ($^{15}N/^{14}N$) by gravity. Hence, each gas isotope was gravity-corrected using Eq. (1):

$$\delta^{18}O_{atm} = \delta^{18}O - 2 \times \delta^{15}N \tag{1}$$

In the same principle, $\delta O_2/N_2$ is affected by a factor of 4 than $\delta^{15}N$; thus it was corrected using Eq. (2), and used to assess the gas loss fractionation:

$$\delta O_2/N_{2,gravcorr} = \delta O_2/N_2 - 4 \times \delta^{15}N \tag{2}$$

Along with gravitational correction, thermal fractionation should also be considered because temperature gradients in the firn column affect the distribution of the isotopes (Severinghaus et al., 1998; Goujon et al., 2003). However, thermal fractionation is typically small in Antarctic ice cores due to the relatively gradual nature of surface climate change (Goujon et al., 2003). Comparing $\delta^{40}Ar/4$ ($^{40}Ar/^{36}Ar$) and $\delta^{15}N$ together allow discrimination of the contribution of thermal and gravity fractionation (Severinghaus et al., 1998). However, for our samples, thermal fractionation could not be considered because $^{36}Ar$ interfered with $^{36}O_2$.

$\delta O_2/N_{2,gravcorr}$ of around $-30$ ‰ indicates that the ice is poorly preserved, and has experienced considerable gas loss either during drilling or storage (Landais et al., 2003). Following Capron et al. (2010), we did not correct for gas loss fractionation in our samples because the $\delta O_2/N_{2,gravcorr}$ values were significantly greater than $-30$ ‰, except for one measurement for sample #301 (Table S1, S2). We did not use the results of #301 when constraining the gas age using $\delta^{18}O_{atm}$. Gravity-corrected $\delta Ar/N_{2,gravcorr}$ ($\delta Ar/N_{2,gravcorr} = \delta Ar/N_2 - 12 \times \delta^{15}N$) is listed in Table S1 and Table S2 but was not used in this study.

## 2.6 $^{81}Kr$ dating

For the $^{81}Kr$ measurement of ice core #23 and TF, 5.3 kg (depth: 711–1040 cm) and 5.4 kg (depth: 798–1192.5 cm) of ice were used, respectively. Air for the $^{81}Kr$ measurement was extracted at SNU using the instrument provided by the University of Science and Technology of China (USTC). The ice was kept in a tank with an O-ring lid and evacuated using a dry scroll pump with a water trap, then melted by immersing the tank in hot water. The released gas was collected in containers and shipped to the USTC for Kr purification and $^{81}Kr$ analysis using Atom Trap Trace Analysis (ATTA). The extraction procedure was described in detail by Tian et al. (2019), and the principle of ATTA is described by Jiang et al. (2012).

The anthropogenic $^{85}Kr$ was measured simultaneously with $^{81}Kr$ to quantify any contamination with modern air. The ice sample details and krypton dating results are presented in Table 1. For both samples, the measured $^{85}Kr$ activity was below the detection limit, hence no correction for contamination with modern air was necessary. For the calculation of the $^{81}Kr$-ages, the changes in the past atmospheric $^{81}Kr$ abundance due to variations in the cosmic ray flux on Earth (Zappala et al., 2020) were considered. The age uncertainty calculation was based on the statistical error of atom counting.

**Table 1. Results of [81]Kr and [85]Kr analysis.** Errors have a 1σ confidence level, whereas upper limits have a 90% confidence level. The [81]Kr age is given with the statistic error due to atom counting. The systematic error of the [81]Kr age is due to the uncertainty in the half-life of [81]Kr (229 ± 11 ka). pMKr = percent modern krypton; dpm cc$^{-1}$ = decay per minute / cubic centimeter STP of Kr. As [85]Kr in modern air in Antarctica is around 65 dpm cc$^{-1}$, the [85]Kr of samples indicates that alteration by modern air is negligible.

| Sample | Mass (kg) | Depth (cm) | [85]Kr (dpm cc$^{-1}$) | [81]Kr (pMKr) | [81]Kr age (ka) | Systematic error (ka) |
|--------|-----------|------------|------------------------|---------------|-----------------|------------------------|
| TF | 5.4 | 798–1192.5 | < 1.2 | 93.5 ± 4.7 | $26^{+15}_{-17}$ | ± 1 |
| #23 | 5.3 | 711–1040 | < 0.7 | 92.4 ± 4.1 | $29^{+14}_{-15}$ | ± 1 |

## 2.7 Development of WD2014 timescale for TALDICE

When constraining the ice age of Larsen ice, the TALDICE record was selected to synchronize Larsen $\delta^{18}O_{ice}$ due to its proximity to the upstream direction of the Larsen BIA. For the TALDICE ice core, we used a timescale that was synchronized to the WAIS Divide (WD) WD2014 chronology (Buizert et al., 2015; Sigl et al., 2016) in the following way: The ice age scales were synchronized using volcanic deposits identified in sulfur/sulfate data from the WD and TALDICE cores (Severi et al., 2012; Buizert et al., 2018). Next, the TALDICE gas-age ice-age difference (Δage) was established empirically by matching abrupt changes in atmospheric $CH_4$ to the WD core; at each match point, this provided one discrete Δage constraint. A dynamical firn densification model based on Herron-Langway firn physics (Herron & Langway, 1980) was then used to interpolate between these empirical Δage constraints in order to obtain a TALDICE gas age scale for all depths (Buizert et al., 2021).

## 3 Results and discussion

### 3.1 Ground penetrating radar (GPR) survey

In the GPR survey, we identified ice layers (or isochrones) in the transect parallel to the ice flow direction (Fig. 3). The dips of the ice layers range from 1° to 6° with a decreasing trend from the upstream to the downstream direction. The ice layers of the radargram were not clearly visible at a depth of < 10 m because of the direct wave signal. We did not observe any stratigraphic folding structure in the ice layers that made age inversion along the ice flow direction in the mid- to downstream areas (from ice cores #23 to #200). Therefore, we expect monotonic and continuous age changes along the ice-flow direction. However, as shown in the dust bands with S- and Z-folds in the upstream area (Fig. 2b), the upstream stratigraphy might be repeated on a scale of tens of meters (from 81w to ice core #23). In addition, the subsurface ice layer in the upstream area (0–800 m from the most upstream side) was not well recognized from the GPR profile (Fig. 3c). It is possible that the noise caused by crevasses, cavities, or cracks could obscure the signals. In addition, accurate data acquisition might have been hindered by antenna tremors or low battery power at severely cold temperatures. The basal topography is well defined from the GPR data; we observed an ice thickness variation of 200–380 m (Fig. 3a, b). The results of the bedrock elevation and ice thickness (Fig.

3a) were obtained using a kriging method, which is a method of interpolation that provides unbiased prediction at unsampled areas (Oliver and Webster, 1990). The ice thickness decreases as ice flows with increasing bedrock elevation, which is a favourable condition for ice to be outcropped.

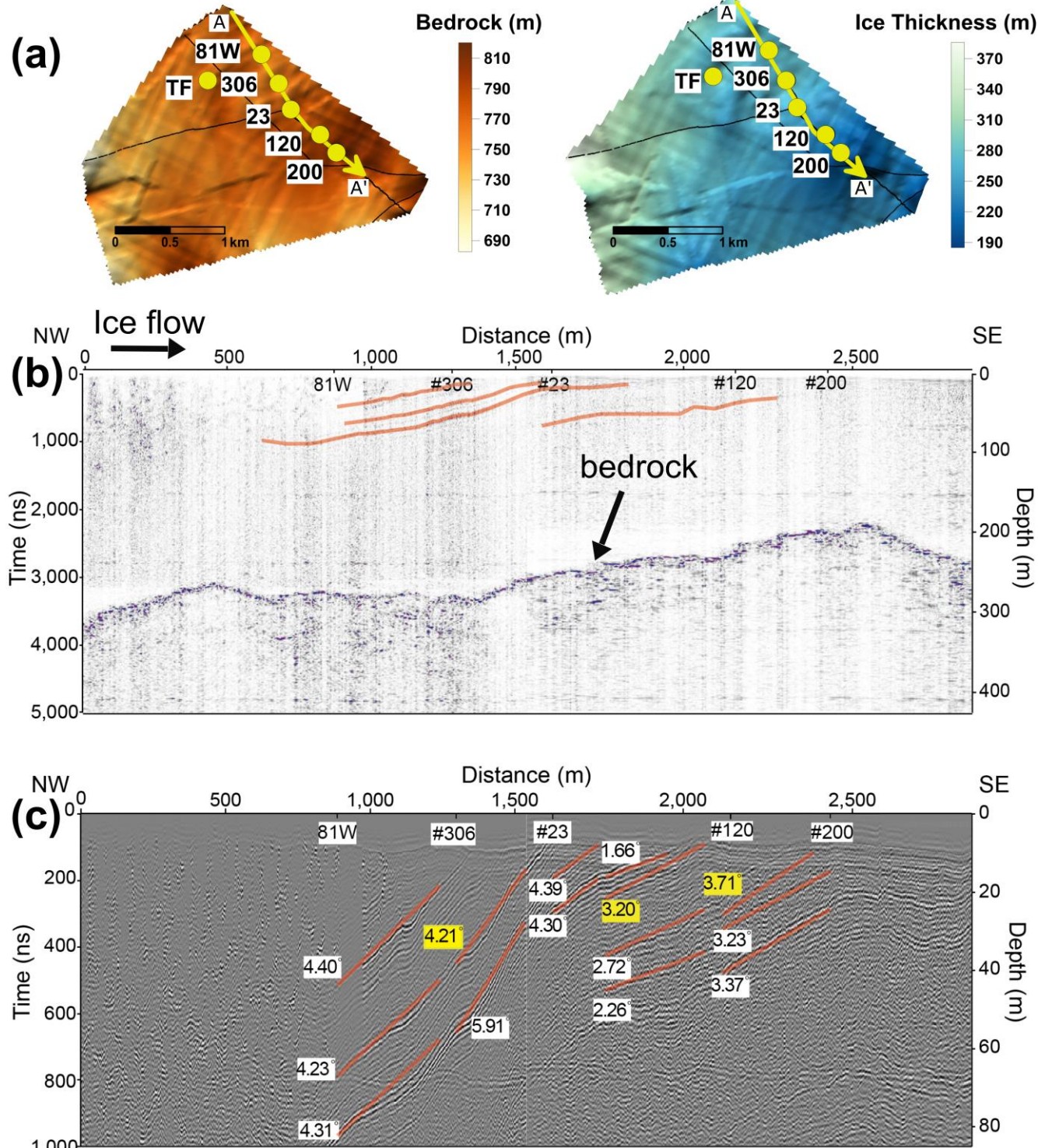

**Figure 3. Ground penetrating radar (GPR) survey profile. (a)** Bedrock elevation and ice thickness of Larsen BIA. **(b)** GPR profile of the transect (AA′ in (a)) through the blue ice field. Identifiable ice layers are indicated in orange lines. Ice layers are not well identified at a distance of < 800 m. **(c)** Enlarged upper 80 m of the GPR profile. Vertical axis is exaggerated and the dip of the ice ranges 1–6°. The average dip used for estimating the ice ages at 1.95 m depth are indicated by yellow boxes (Appendix C).

## 3.2 Stable water isotopes

Stable water isotopes in ice record surface temperatures in the past at snow deposition sites (Jouzel et al., 1997). The $\delta^{18}O_{ice}$ and $\delta^2H_{ice}$ records of Larsen BIA are presented in Table S5, S6, and S7. The horizontal $\delta^2H_{ice}$ result has a distinct local minimum around a horizontal distance of 600–800 m, indicating a transient cold event (Fig. 4b). The $\delta^2H_{ice}$ of the transient cold event plunges by approximately 70 ‰, heading to the upstream ice. The most downstream sample, ice core #200, had a $\delta^2H_{ice}$ of −369 ‰. Conversely, the most upstream sample, surface ice 81w, had a $\delta^2H_{ice}$ of −245 ‰. Ice core #23 was located in the middle of the transect, and the vertical $\delta^2H_{ice}$ profile showed a range of −353 ‰ to −291 ‰. It appears that the water isotope values from the Larsen BIA are more scattered over a wider range than other published ice core records (Petit et al., 1999; EPICA Community Members, 2004; Stenni et al., 2011). A highly variable $\delta^{18}O_{ice}$ has also been reported in the Taylor Glacier (Baggenstos et al., 2018; Menking et al., 2019). Severe scattering might indicate that the accumulation zone (original source of ice) of Larsen BIA might have experienced more variability in temperature and/or the vapour source (variability of atmospheric conditions) than other sites. Deuterium-excess ($d = \delta^2H_{ice} − 8 \times \delta^{18}O_{ice}$) shows a wide range (5.40 to −3.89 ‰, Table S5) from the entire near-surface ice samples. The negative d-excess likely indicates that isotopic fractionation is attributed to the sublimation of ice in the accumulation zone (Hu et al., 2021). Negative d-excess values were also observed in the Allan Hills BIA (Hu et al., 2021). Meanwhile, sublimation can deplete $^{16}O$ and $^1H$ in ice and make the isotopic ratios ($\delta^{18}O_{ice}$ and $\delta^2H_{ice}$) enriched. Thus, the wide range and negative value of d-excess results indicate that stable water isotopes are not proper proxies for the changes in temperature and vapour sources.

To match the two $\delta^2H_{ice}$ profiles (measurement of ice core #23 and horizontal measurement of near-surface ice), the depth of ice core #23 was converted to the horizontal distances by pinpointing the deepest result of #23 to the result of ice core #104 and conducting linear interpolation. The two $\delta^2H_{ice}$ profiles had an $r^2$ value of 0.86 (p < 0.001) (Fig. 4b). The similarity of $\delta^2H_{ice}$ between the vertical and horizontal measurements demonstrates that the ice stratigraphy was not disturbed. The average offset of the values of $\delta^2H_{ice}$ between vertical ice core #23 and the horizontal record is 9.8 ‰ and 0.6 ‰ for $\delta^{18}O_{ice}$. We assume that the average offset is the uncertainty (1σ) of stable water isotope values of the Larsen ice. As the ~10 m long #23 core covers ages that correspond to a surface distance of ~117 m, we calculated the average dip of the ice layer of 4.96°, which is comparable with the average dip derived from the GPR profile (Fig. 3c). This depth/distance relationship is also supported by the comparison of gas isotope values ($\delta^{18}O_{atm}$ and $\delta^{15}N$-$N_2$) from ice core #23 to the records from the horizontal measurement at a depth of 1.95 m (Fig. A2).

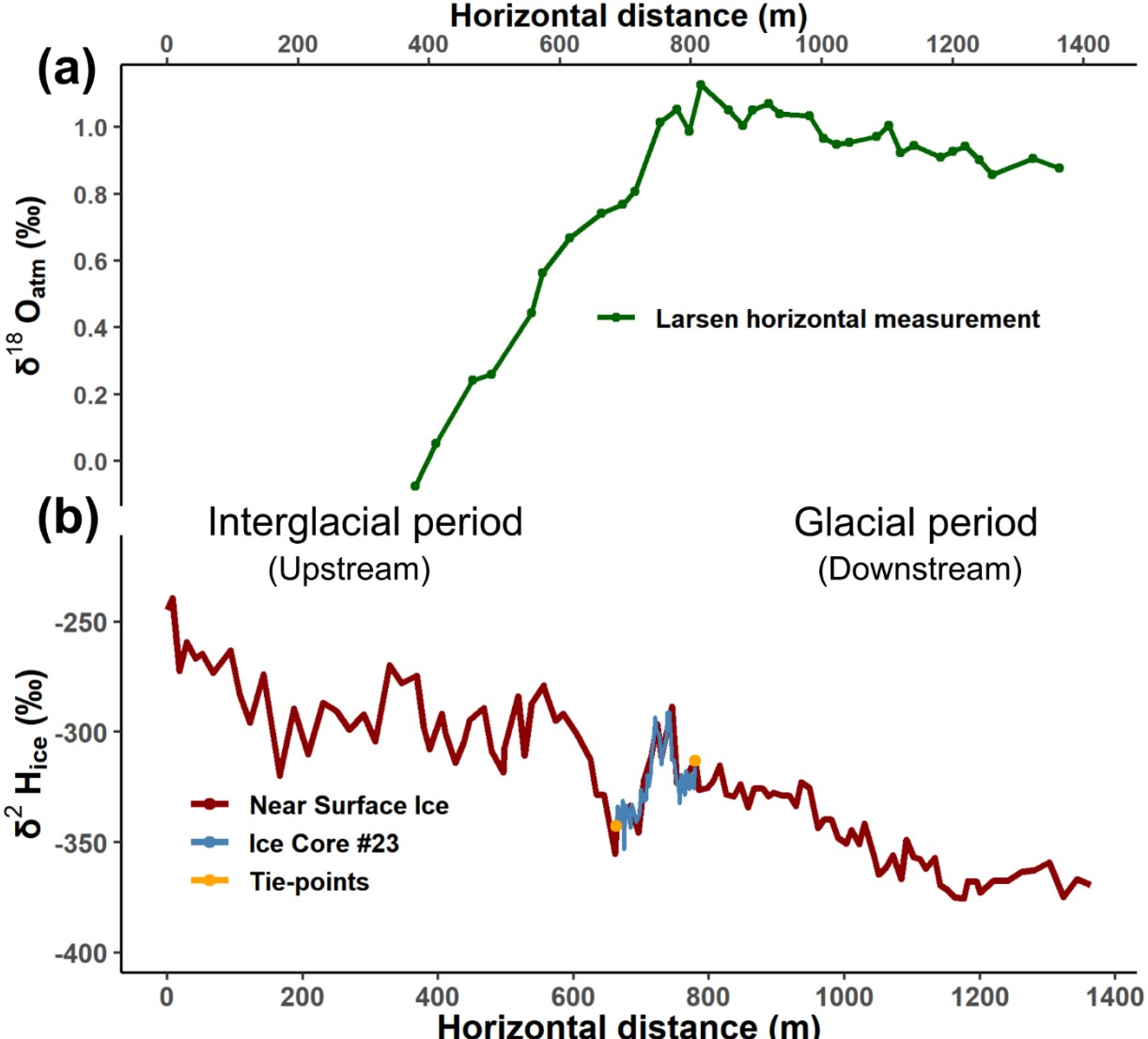

**Figure 4. $\delta^{18}O_{atm}$ and $\delta^2H_{ice}$ records from Larsen BIA. (a)** $\delta^{18}O_{atm}$ from each ice core at a 1.95 m depth (horizontal measurement). The $\delta^{18}O_{atm}$ of the horizontal measurement was gravity-corrected using Eq. (1). **(b)** $\delta^2H_{ice}$ from near-surface ice (~5–10 cm depth surface ice samples and 10–30 cm depth ice core samples) and ice core #23. The surface ice record was matched with core #23 using two tie-points (orange dots) and show a $r^2$ value of 0.86 (p < 0.001).

### 3.3 Analysis of gas entrapped in the ice

#### 3.3.1 CH₄ and CO₂ mixing ratios

The measurement results of the $CH_4$ and $CO_2$ concentrations for the vertical cores are presented in Fig. A1. The shallow ice cores (#306, #120, #201) show that greenhouse gases are significantly altered for the top 2 m, showing out-of-range values of the range of natural greenhouse gas concentrations during the last 800 ka BP: 340–800 ppb for $CH_4$ and 180–300 ppm for $CO_2$.

The $CH_4$ records of #23 also fluctuate significantly at 0–4.6 m depth but settle down at > 4.6 m. $CO_2$ records of #23, in contrast, gradually decrease and become steady at a depth of > 4.6 m. A comparison of the results from NIPR to SNU with ice core #23 shows that the difference in the concentration decreases significantly at depths of > 4 m; large differences (30–140 ppb) of
$CH_4$ at 0–4 m depths, but decreased considerably (5–10 ppb) at a depth of > 4 m; the $CO_2$ difference was 10–20 ppm at < 4 m, but decreased to 2–10 ppm at a depth of > 4 m (Table S2 and S3). The $CO_2$ mixing ratio of the TF core, reported by Jang et al. (2017), showed that $CO_2$ was scattered even in the deeper part (> 4 m) of the ice core. We speculate that this scattering is due to complicated ice stratigraphy since we observed several folding structures identified by dust bands near the TF core site (Fig. 2b). Altered $CH_4$ in near-surface samples have also been shown in Pakitsoq, Taylor Glacier, and Elephant Moraine (Petrenko
et al., 2006; Baggenstos et al., 2017). The depth at which greenhouse gas concentrations stabilized appears to be inversely proportional to mean annual temperature (Table 2).

**Table 2. Depth of unaltered greenhouse gas compositions and mean annual temperature of BIAs.**

| Site | Depth of unaltered greenhouse gas composition (m) | Mean annual temperature (°C) | Reference for depth of unaltered air | Reference for mean annual temperature |
|---|---|---|---|---|
| Elephant Moraine Texas Bowl | > 10[*] | -30.3 | This study (Fig. A1) | KOPRI AWS (76.27° S, 156.71° E) |
| Allan Hills | > 7–10[§] | -31[†] | Spaulding et al. (2013) | Delisle and Sievers (1991) |
| Larsen Glacier | > 4.6 | -24.4 | This study (Fig. A1) | Antarctic Meteo-Climatological Observatory |
| Taylor Glacier | > 4 | -18[‡] | Baggenstos et al. (2017) | United States Antarctic Program |
| Pakitsoq | > 0.3 | -5.4[¶] | Petrenko et al. (2006) | Climate-data.org |

[*]Depth of unaltered greenhouse gas compositions from Elephant Moraine Texas Bowl remains uncertain due to the lack of data at depth of > 10 m. Mean annual temperature of Elephant Moraine: provided by KOPRI's automatic weather station (AWS) record of year 2020 and
2021. [†]Allan Hills: not provided by an AWS but by stable water isotopes; [§]deduced by vertical profile of $\delta^{15}N$-$N_2$ and $\delta^{18}O_{atm}$ values in Allan Hills. [‡]Taylor Glacier: assumed to be comparable with the mean annual temperature of nearby McMurdo station (~100 km away). [¶]Pakitsoq: assumed to be comparable with the mean annual temperature of the nearby town of Ilulissat (~ 40 km away).

### 3.3.2 $\delta^{15}N$-$N_2$ and $\delta^{18}O_{atm}$

As $\delta^{15}N$-$N_2$ and $\delta^{18}O_{atm}$ were measured at very shallow depths (~1.95 m), we compared the horizontal results with the
vertical distribution of ice core #23. The depths of ice core #23 were converted to horizontal distances at a depth of 1.95 m (Fig. A2) using Eq. (3):

Horizontal distance in meter (at 1.95 m depth) = [{(depth of ice core #23) − 1.95}/tan(4.96°)] + 663       (3)

We added 663 m because ice core #23 was located 663 m from the most upstream sampling site. The average dip of the ice layer (4.96°) was calculated by matching the $\delta^2H_{ice}$ of ice core #23 with the horizontal records from the neighbouring cores
(Fig. 4b). The $\delta^{15}N$-$N_2$ and $\delta^{18}O_{atm}$ values obtained from the horizontal record at 1.95 m are comparable to those from ice core #23, except those from the very shallow depths of < 0.5 m (Fig. A2), confirming that the gas isotope ratios are generally reliable at a depth of 1.95 m. We estimated the uncertainty of $\delta^{15}N$-$N_2$ and $\delta^{18}O_{atm}$ values as ±0.05 ‰ (1σ) because the offsets

between #23 and the horizontal record were approximately 0.05 ‰. Petrenko et al. (2006) also reported that as long as the ice is not affected by surface ice melting, nitrogen and oxygen gas isotopes are not altered even at a depth of 0.3 to 0.4 m at the

western Greenland ice. In contrast, $\delta^{15}N$-$N_2$ and $\delta^{18}O_{atm}$ values in the Allan Hills are stabilized below 7–10 m (Spaulding et al., 2013). No significant gaps, discontinuities, or anomalies were found within the results from horizontal measurements of $\delta^{18}O_{atm}$, which indicate no significant stratigraphic disturbance (Fig. 4a).

### 3.4 Glacial termination identification in Larsen ice

  The increase in $\delta^2H_{ice}$ and decrease in $\delta^{18}O_{atm}$ from downstream to upstream ice shows that the atmospheric conditions

changed to a warmer climate (Fig. 4). In particular, the $\delta^{18}O_{atm}$ values of the Larsen ice (1.126 ‰ to −0.075 ‰) reveal a typical glacial termination period. The maximum $\delta^{18}O_{atm}$ value of > 1 ‰ implies that the ice is younger than 800 ka BP because the glacial climate conditions became more extreme following MPT (Lisiecki and Raymo, 2005; Elderfield et al., 2012; Chalk et al., 2017); it is likely that the maximum $\delta^{18}O_{atm}$ values were < 1.0 ‰ during the pre-MPT glacial periods (Yan et al., 2019). The EPICA Dome C (EDC) record shows that Termination I, II, IV, V and VII are the only terminations that have both values

of negative and > 1.0 ‰ during the last 800 ka BP (Landais et al., 2013; Extier et al., 2018a) (Fig. 5). Hence, we exclude Termination III and VI for the Larsen BIA ages. Among the candidates, Termination V should also be excluded because the maximum $\delta^{18}O_{atm}$ value in the EDC record is ~1.4 ‰, which is significantly higher than that of the Larsen ice. The $\delta^2H_{ice}$ decrease in the middle of the glacial termination in the Larsen BIA (Fig. 4b) is similar to that during the Antarctic Cold Reversal (ACR, 12.7–14.6 ka BP), which is a distinct feature only during Termination I among the candidate terminations (i.e.,

Termination I, II, IV and VII) as observed in the EDC $\delta^2H_{ice}$ record (Fig. 6). To confirm the age of the Larsen ice, we compared the $\delta^{18}O_{atm}$–$CH_4$ relationship with the ice from core #23 at depths of 4.6–10 m with the EDC and WAIS Divide (Fig. 7), and found that the Larsen $\delta^{18}O_{atm}$–$CH_4$ distribution was well matched only at Termination I. The $\delta^{18}O_{atm}$ of Larsen has a 0.05 ‰ offset with that of the WAIS Divide, which is in line with our observation of ~0.05 ‰ offset between core #23 and horizontal measurements at a depth of > 1.95 m (Fig. A2). The offset may also come from the age difference because some $\delta^{18}O_{atm}$ records

are estimated by linear interpolation due to the lack of a $\delta^{18}O_{atm}$ record corresponding to the $CH_4$ of core #23. The $\delta^{18}O_{atm}$–$CO_2$ and $CO_2$–$CH_4$ relationships do not clearly match with those of Termination I in the EDC and WAIS Divide records. The Larsen ice showed a higher $CO_2$ concentration of 10–20 ppm (Fig. A3, Fig. A4). $CO_2$ concentration warrants further investigations. Finally, we confirmed the ages of the Larsen ice with [81]Kr dating, indicating 9–41 and 14–43 ka for ice from the TF and #23 cores, respectively (Table 1), and concluded that Larsen ice covers the Last Glacial Termination (LGT, T1).

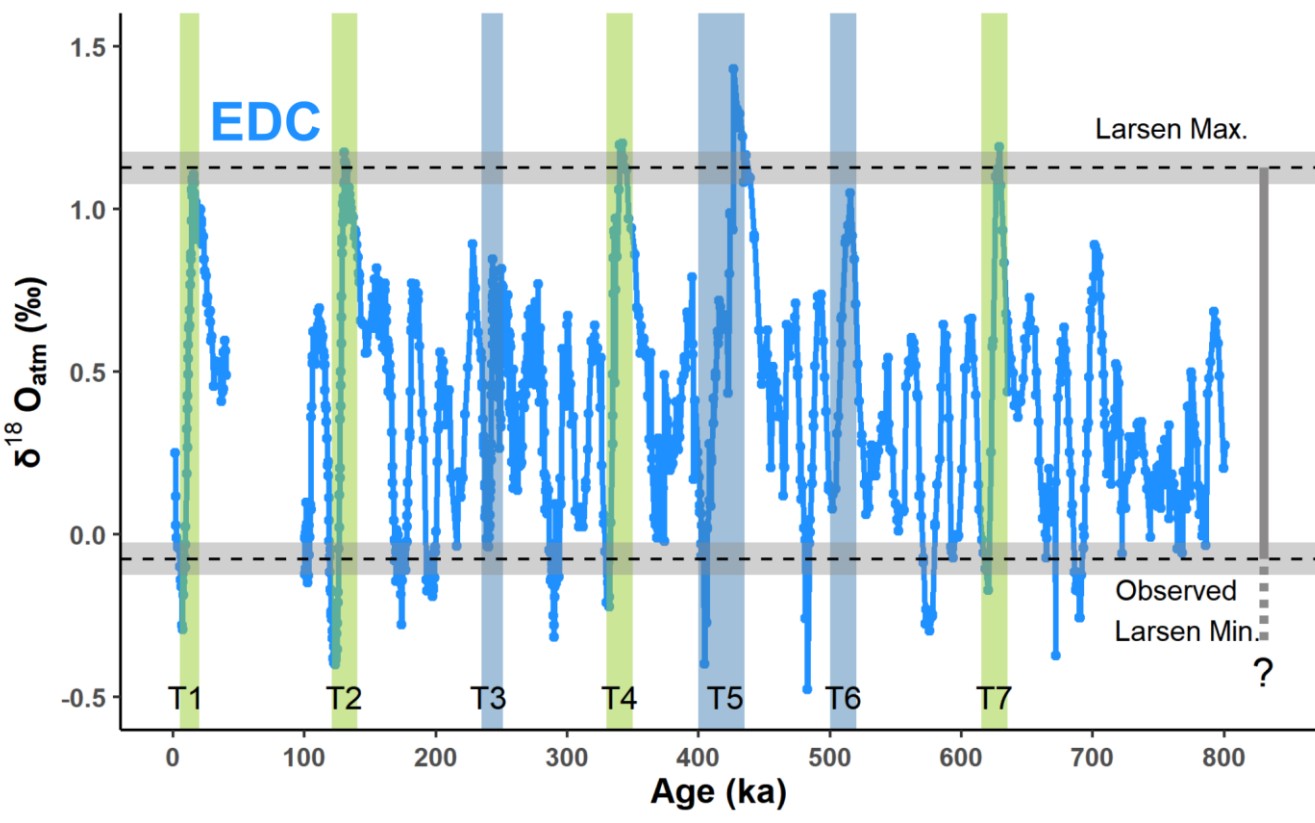


**Figure 5. Comparison of Larsen δ¹⁸O_atm with EDC record.** $\delta^{18}O_{atm}$ records are from Landais et al. (2013) for 0–40 ka and Extier et al. (2018a) for 100–800 ka, respectively. Green vertical bars represent the candidate age intervals for Larsen blue ice; T, Termination. Observed Larsen $\delta^{18}O_{atm}$ value is shown as a vertical grey bar. Horizontal grey bars with dash lines represent the uncertainty of the measured $\delta^{18}O_{atm}$ from Larsen BIA (Fig. A2).


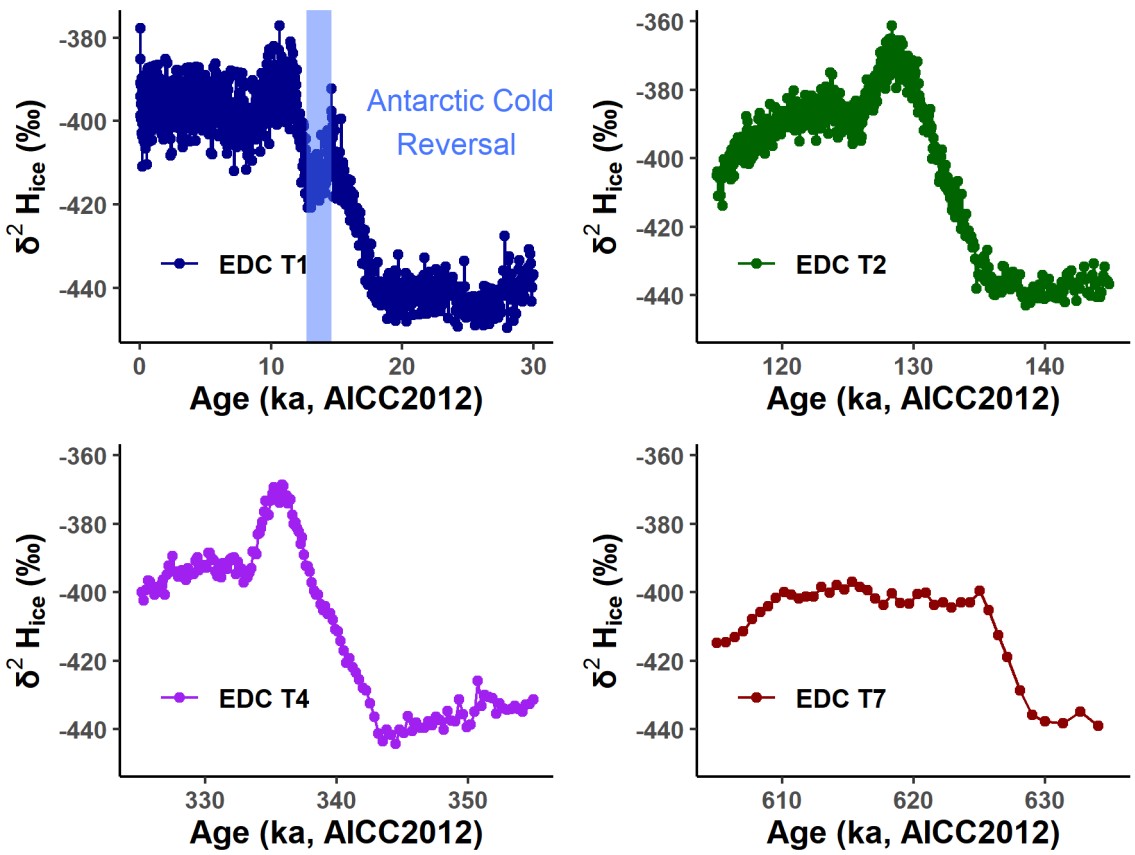

**Figure 6. δ²H$_{ice}$ records from EDC.** δ²H$_{ice}$ records for T1, T2, T4, and T7 on AICC2012 scale are from Bazin et al. (2013b). The blue vertical bar represents the time interval of the ACR (12.7–14.6 ka BP).

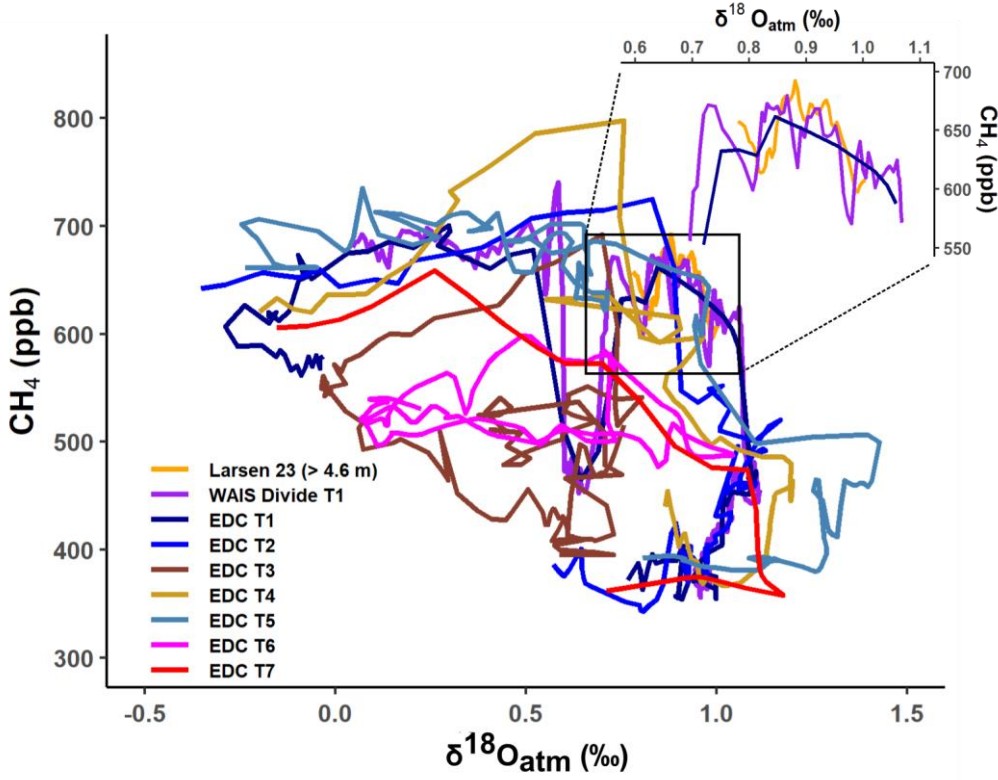

**Figure 7. Comparison of $\delta^{18}O_{atm}$-CH$_4$ relationship in the Larsen ice core #23 with existing records during glacial Terminations.** $\delta^{18}O_{atm}$ and CH$_4$ data of WAIS Divide are from Severinghaus (2015) and Rhodes et al. (2017), respectively. $\delta^{18}O_{atm}$ of EDC is from Landais et al. (2013) and Extier et al. (2018a). CH$_4$ record of EDC is from Bazin et al. (2013d). The area within the black box is magnified to compare the Larsen #23 record with those from WAIS Divide T1 and EDC T1.

### 3.5 Gas and ice ages of Larsen ice

The CH$_4$ was severely altered at a depth of 1.95 m and the available CH$_4$ record was only from ice core #23 (> 4.6 m). Therefore, we developed the tentative gas age by correlating the horizontal $\delta^{18}O_{atm}$ value to the WAIS Divide record on the WD2014 timescale. Then, fine correlation was conducted with CH$_4$ record from the core #23, which corresponds to 13.3–14.1 ka BP at depths of 4.6–10.4 m (Fig. 8d). We used the high-resolution records of CH$_4$ from both Larsen core #23 and WAIS Divide ice, which greatly improves the precision of age construction. The following paragraph describes this process in detail.

Spline curves were created for both WAIS Divide and Larsen $\delta^{18}O_{atm}$ records to reduce artefacts from insufficient sampling resolution and/or ice quality at a 1.95 m depth of the Larsen ice (Fig. 8). The spline curve was drawn after interpolating the original records at 5 m horizontal distance intervals. Three points were selected, which divide the upstream part of the Larsen ice into three equal parts to tie with those in the WAIS Divide record (pink dots in Fig. 8a and 8b). For the downstream side of the Larsen ice, we selected a local maximum and a minimum as the tie points (purple dots in Fig. 8a and 8b) because the slope of the $\delta^{18}O_{atm}$ spline curve is small. When the horizontal distance was > 1,280 m, the age was constrained by extrapolation

using the age/distance relationship at 18–22 ka. Based on the tentative horizontal gas ages and the depth/distance relationship of core #23 (obtained from Fig. 4b), we determined the gas age of core #23 (~10 m vertical ice core). A small offset (53.6 ± 38.5 yr) existed between the $CH_4$ record of core #23 (> 4.6 m) and the WAIS Divide record (offset between the red and dark blue lines in Fig. 8d). To eliminate this gap, four tie-points (orange dots) were chosen and interpolated (light blue line). The gas age, corresponding to 13.3–14.1 ka BP was obtained more accurately via $CH_4$ correlation. As a result, the gas age for the

Larsen ice at 1.95 m depth was estimated to be 9.2–23.4 ka BP on WAIS Divide chronology in 2014 (WD2014). The ages on AICC2012 scale were estimated to be 9.4–23.4 ka BP using the depth and the WD2014 age of EDC (Bazin et al., 2013a, Buizert et al., 2021). The age/distance relationship was reasonable (Fig. A7), showing no abrupt change, as supported by the gradual change in the ice layer dip (Fig. 3c).

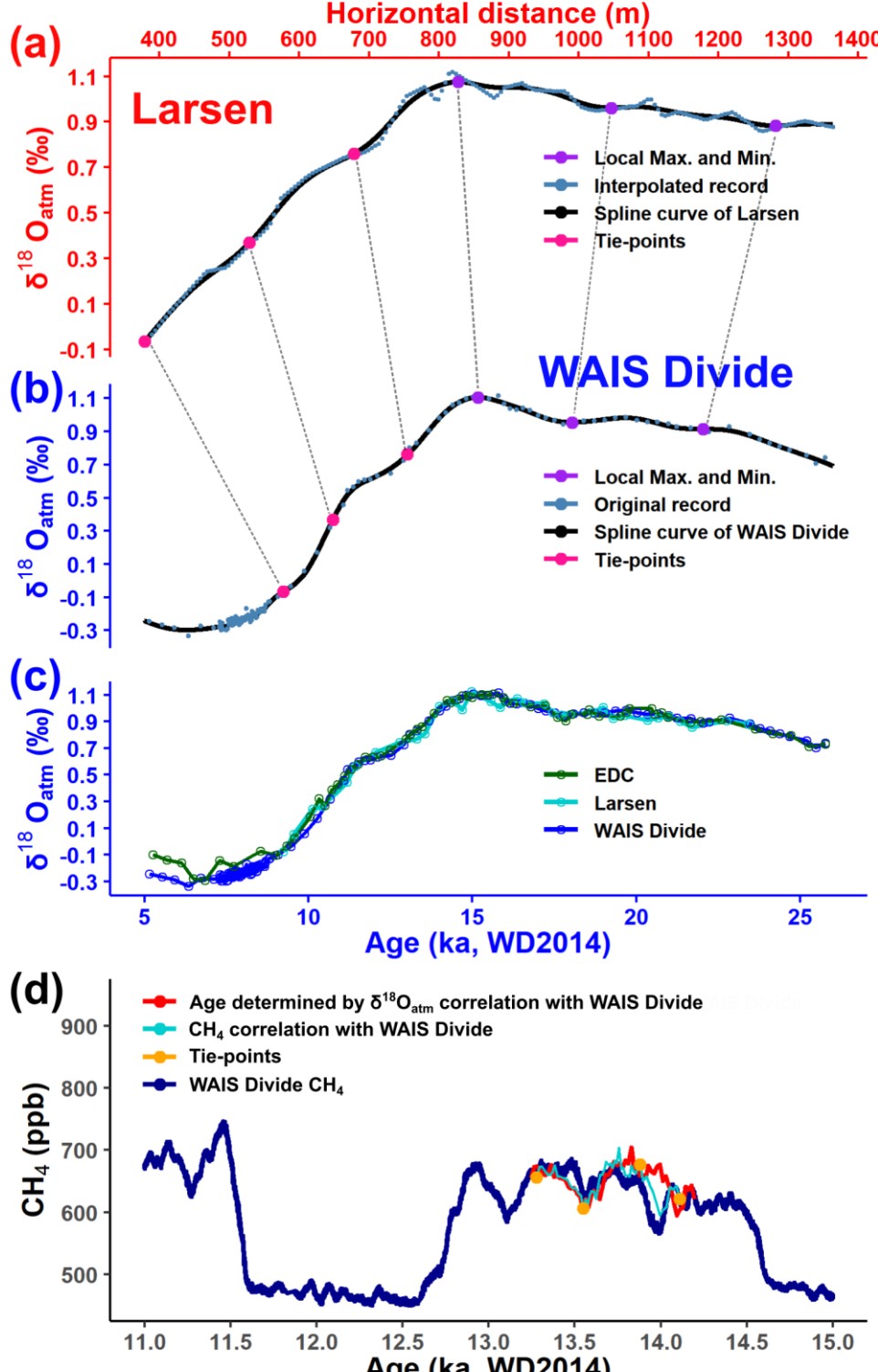

**Figure 8. Correlation of Larsen δ¹⁸O_atm and CH₄ records with WAIS Divide. (a)** 5 m interpolated δ¹⁸O_atm value from Larsen BIA with a spline curve. **(b)** δ¹⁸O_atm record from WAIS Divide with a spline curve (Severinghaus et al., 2015). Six tie-points were used to correlate each other. **(c)** Comparison of synchronized Larsen δ¹⁸O_atm with EDC (Landais et al., 2013) and WAIS Divide records. **(d)** Comparison of CH₄ record from Larsen #23 (> 4.6 m depth) with WAIS Divide (Rhodes et al., 2017). Tentative gas age determined by δ¹⁸O_atm correlation with WAIS Divide is tuned by correlating CH₄ record using four tie-points. WD2014 timescale for WAIS Divide and EDC is from Sigl et 415 al. (2016) and Buizert et al. (2021), respectively.

The ice ages of the Larsen ice were determined based on the δ¹⁸O_ice correlation with the TALDICE record (Fig. 9). TALDICE was selected since the coring site is close to the Larsen BIA (~240 km apart) and the direction of the TALDICE site from the Larsen BIA is similar to the ice flow direction of the Larsen ice (Rignot et al., 2011; Mouginot et al., 2012). It is likely that the trend of surface temperature changes in the snow accumulation zone of the Larsen BIA is comparable to that of the TALDICE 420 site. Spline curves were constructed for the horizontal measurement of the surface ice. As described above, we interpolated the original δ¹⁸O_ice records at 5 m intervals to avoid bias in the spline curve of the δ¹⁸O_ice record. The correlation was established by visually selecting a similar inflection point. Then, to validate the ice age, we estimated the ice age at a 1.95 m depth (Appendix C for more information) and calculated the Δage; Δage for Larsen ice is defined as (ice age) − (gas age) at a depth of 1.95 m.

First, eight tie-points were selected and linearly interpolated. In this case, Δage showed a minimum value at around 17.5 ka (Fig. A6), which is close to the period of the Last Glacial Maximum (LGM). Since Δage increases when the climate is colder (Schwander et al., 1997), we assumed that either the ice age is undervalued or the gas age is overvalued. Because the δ¹⁸O_atm value of the Larsen ice is comparable with the EDC and WAIS Divide records, the gas age seems to be reasonable and not likely to be overvalued (Fig. 8). Hence, we concluded that the ice age should be revised. In addition, the age/distance 430 relationship of ice shows a rapid age increase near ~1,000 m (Fig. A7), which is not supported by the average dip of the ice layer deduced from the GPR profile (Fig. 3). As the dip decreases to the downstream ice, no abrupt increase in the slope of the age/distance relationship is expected. However, this is the case only when we assume no abrupt change in the snow accumulation rate.

After excluding one tie-point (indicated by the red dotted line in Fig. 9), the Δage of Larsen ice shows a maximum value at 435 approximately 17.5 ka BP, which appears more appropriate (Fig. A6). For records with horizontal distances less than 107 m and greater than 1,310 m, extrapolation was conducted to estimate the ice age using the age/distance relationship at 7–11 ka BP and 18–23.8 ka BP, respectively. In conclusion, the ice age was estimated to be 5.6–24.7 ka BP (WD2014) for the surface ice. However, as noted above, local age inversion may have occurred in the upstream ice areas. Hence, the age constraint for upstream ice should be considered cautiously. The ice age calculated for a 1.95 m depth also seems reliable because the water 440 isotope record is comparable with the surface ice result when plotted with the ice age (Fig. 9c). Note that ice age synchronization via water isotopes is challenging in climatically stable periods such as the LGM; measurements of dust and ice chemistry could be applied in the future to refine the ice chronology presented here.

.

The established horizontal ice chronology shows that temporal resolutions of 10 yr m⁻¹ and 17 yr m⁻¹ are available for the 445 Holocene (5.6–12 ka BP) and for the last deglaciation (12–24.6 ka BP), respectively, at the surface ice from Larsen BIA. These

estimated resolutions are higher than those of the deep ice core TALDICE covering this time interval (~18 yr m$^{-1}$) (Masson-Delmotte et al., 2011). Assuming that the average age/depth relationship of ice core #23 (~189 yr m$^{-1}$) is maintained throughout the ice, the ice age may then reach ~60 ka near the bedrock (~240 m of ice), and thus, ice may be obtained at least ~60 ka via shallow coring at the Larsen BIA.

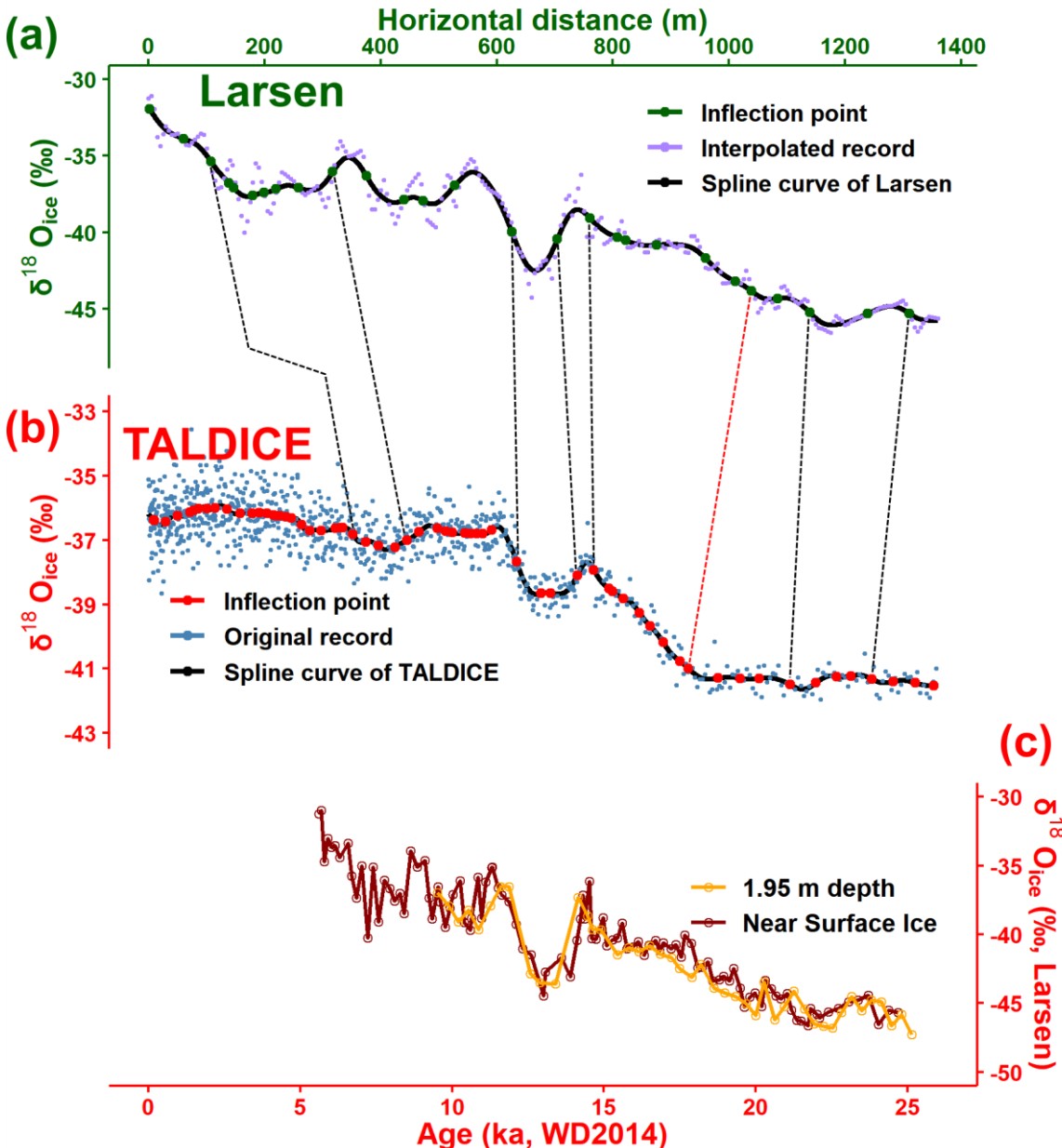

450

**Figure 9. Correlation of Larsen δ$^{18}$O$_{ice}$ with TALDICE record. (a)** 5 m interpolated δ$^{18}$O$_{ice}$ value from Larsen BIA with a spline curve. **(b)** δ$^{18}$O$_{ice}$ record from TALDICE with a spline curve (Stenni et al., 2011, Bazin et al., 2013c). The ice age of the surface ice from Larsen was estimated by correlating 7 inflection points of the spline curve (indicated by black dotted lines) with TALDICE. **(c)** Comparison of near-surface δ$^{18}$O$_{ice}$ with those in 1.95 m depth at Larsen BIA.

## 3.6 Age uncertainty

### 3.6.1 Gas age uncertainty

There are two types of uncertainty to consider: (1) the relative Larsen gas age uncertainties to the WAIS Divide (WD) gas age, and (2) the absolute WD gas age uncertainty itself. To assess the relative Larsen gas age uncertainty to WD gas age, we applied a Monte Carlo simulation running the model 10000 times. The model is described in detail in the following paragraph.

First, to assign relative Larsen gas age uncertainties of the points that were pin-pointed to WD (pink and purple dots in Fig. 8a, b), analytical uncertainty should be defined. The analytical uncertainty of $\delta^{18}O_{atm}$ from Larsen is assumed to be ±0.05 ‰, as discussed in Sect. 3.3.2. The analytical uncertainty of $\delta^{18}O_{atm}$ from WD was ±0.006 ‰ (Severinghaus et al., 2015). Because the analytical uncertainty of $\delta^{18}O_{atm}$ from WD is about one order of magnitude lower than Larsen's uncertainty, we assume the total analytical uncertainty to be ±0.05 ‰. Then, the gas age for the pink dots (Fig. 8a, b) was assigned 10,000 times using a Monte Carlo simulation considering the analytical uncertainty. The standard deviation of the assigned ages were used for the relative gas age uncertainty.

The relative Larsen gas age uncertainty estimation process for the older tie-points (purple dots in Fig. 8a, b) is more complicated. We repeatedly produced the spline curve 10000 times using the Monte Carlo approach and found the location (horizontal distance) of the local maximum and local minima. The simulation results were rejected when there was no or only one local minimum. This Monte Carlo result and its uncertainties were used to detect outliers (> 2σ). The process was iterated until no outliers were detected. Using the location (horizontal distance) uncertainty of the local maximum and local minima, we then estimated the relative Larsen gas age uncertainty to WD gas age using the Monte Carlo simulation.

The absolute WD gas age uncertainty itself should also be considered (Sigl et al., 2019). Because the relative Larsen gas age uncertainty to WD gas age and the absolute WD gas age uncertainty are independent, the total Larsen gas age uncertainty can be calculated using Eq. (4):

$$\sigma_{total} = \sqrt{\sigma_{abs.}^2 + \sigma_{rel.}^2} \tag{4}$$

The uncertainties are presented in Table 3. The gas age uncertainty between the tie-points was determined by linear interpolation, and the uncertainty located outside the last tie-point was linearly extrapolated using the uncertainty/age relationship of 18–22 ka BP.

**Table 3. Result of gas age uncertainties.** $\sigma_{rel.}$, $\sigma_{abs.}$, and $\sigma_{total}$ represent relative Larsen gas age uncertainty to WD gas age, absolute WD gas age uncertainty, and total uncertainty of Larsen gas age, respectively.

| Gas age (ka) | $\sigma_{rel.}$ (ka) | $\sigma_{abs.}$ (ka) | $\sigma_{total}$ (ka) |
|---|---|---|---|
| 9.24 | 0.324 | 0.055 | 0.329 |
| 10.74 | 0.130 | 0.062 | 0.144 |
| 13.02 | 0.274 | 0.117 | 0.298 |
| 15.17 | 0.084 | 0.158 | 0.179 |
| 18.05 | 0.194 | 0.214 | 0.289 |

| 22.04 | 0.254 | 0.257 | 0.361 |

### 3.6.2 Ice age uncertainty

Similar to the Larsen gas age uncertainty, the Larsen ice age uncertainty consists of (1) the relative Larsen ice age uncertainty to TALDICE ice age, and (2) the absolute TALDICE ice age uncertainty itself. The total analytical uncertainty of $\delta^{18}O_{ice}$ is

assumed to be ±0.6 ‰, the same as the $\delta^{18}O_{ice}$ uncertainty of Larsen ice (Sect. 3.2) because the analytical uncertainty of $\delta^{18}O_{ice}$ for TALDICE (±0.07 ‰) is negligible (Stenni et al., 2011). The method to constrain the relative Larsen ice age uncertainty to the TALDICE ice age is similar to the case of estimating the gas age uncertainty of the older tie-points (purple dots in Fig. 8a, b). In contrast, in this case, we found the location (horizontal distance) of the inflection points, not the local maxima or minima. The relative uncertainty can also be derived from when choosing the tie-points in TALDICE. In addition, the ice age

uncertainty for a depth of 1.95 m should be larger than the uncertainty we provide because it was estimated using the average dip of the ice layer (Appendix C). Therefore, the relative Larsen ice age uncertainty provided here is a lower limit.

The absolute TALDICE ice age uncertainty itself should also be considered, where the uncertainty is calculated by quadratically combining the absolute WD ice age uncertainty and the volcanic synchronization uncertainty. Then, quadratically combing the absolute TALDICE ice age uncertainty and the relative Larsen ice age uncertainty to the TALDICE ice age (Eq.

(4)) provides the total Larsen ice age uncertainty (Table 4). The ice age uncertainty between the tie-points was determined by linear interpolation, and the uncertainty located outside the last tie-point was linearly extrapolated using the uncertainty/age relationship of 18–23.8 ka BP.

**Table 4. Result of ice age uncertainties.** $\sigma_{rel.}$, $\sigma_{abs.}$, and $\sigma_{total}$ represent relative Larsen ice age uncertainty to TALDICE ice age, absolute TALDICE ice age uncertainty, and total uncertainty of Larsen ice age, respectively.

| Ice age (ka) | $\sigma_{rel.}$ (ka) | $\sigma_{abs.}$ (ka) | $\sigma_{total}$ (ka) |
|---|---|---|---|
| 6.73 | 0.023 | 0.030 | 0.038 |
| 8.51 | 0.014 | 0.040 | 0.043 |
| 12.12 | 0.044 | 0.084 | 0.095 |
| 14.12 | 0.023 | 0.135 | 0.137 |
| 14.67 | 0.041 | 0.147 | 0.152 |
| 21.14 | 0.053 | 0.211 | 0.218 |
| 23.83 | 0.060 | 0.238 | 0.246 |

### 3.6.3 Δage uncertainty

To estimate the Larsen Δage uncertainty, the relative Larsen gas age uncertainty and relative Larsen ice age uncertainty should be on the same ice core. Here, we used the relative Larsen gas age and ice age uncertainties to the WD gas age and ice age, respectively. We assumed that the Larsen Δage uncertainty consists of (1) the relative Larsen gas age uncertainty to WD gas age, (2) the relative Larsen ice age uncertainty to WD ice age, and (3) the Δage uncertainty of WD. Quadratically combining

these three components provides the total Larsen Δage uncertainty, similar to Eq. (4). The relative Larsen ice age uncertainty

to TALDICE ice age and the relative TALDICE ice age uncertainty to WD ice age were quadratically combined to estimate the relative Larsen ice age uncertainty to WD ice age. Volcanic synchronization uncertainty is used for the relative TALDICE ice age uncertainty to the WD ice age. The Larsen Δage uncertainty is presented in Table S8.

### 3.7 Estimation of past surface temperature and accumulation rate

As discussed in Sect. 3.2, $\delta^{18}O_{ice}$ and $\delta^2H_{ice}$ might have been enriched during the sublimation of ice. Therefore, it is inadequate to use stable water isotopes to estimate past surface temperatures. Instead, the analytical framework developed by Buizert (2021) allows us to estimate the past surface temperature using the $\delta^{15}N$-$N_2$ and Δage. The methods for estimating the surface temperature and accumulation rate are described in detail in the Supplement (Text S1). The results are presented in Table S9 and Fig. 10. The accumulation rates in TALDICE, EDC, Taylor Dome, and Taylor Glacier are also shown for

comparison (Veres et al., 2013; Baggenstos et al., 2018). The youngest three reconstructed values of Larsen BIA were rejected (red line in Fig. 10) due to the large relative uncertainty in Δage (including the possibility of negative Δage), and the implied 10.5 °C increase in just ~1,200 years. This might be due to a low bias in the reconstructed ice age, which leads to a lower Δage. Hence, the ice age in this part should be more strongly constrained by using dust concentrations in a future study.

During the last deglaciation, the surface temperature increased by approximately 15 ± 5 °C (from 24.3 ka BP to 10.6 ka BP),

which is greater than those for any other ice core sites including nearby Taylor Dome where a recent reconstruction suggests 7.1 ± 2 °C of deglacial warming (Buizert et al., 2021). However, the interpretation should be cautious because of the large uncertainty. The large magnitude of the reconstructed temperature change is a consequence of Δage, which is around 300 % larger during the Last Glacial Maximum (LGM) than during the Holocene in our reconstruction. By comparison, most Antarctic sites show only a 60 % to 120 % Δage increase (Buizert et al., 2021). While the Larsen BIA is currently close to the

open ocean, during the LGM, the Antarctic ice sheet reached the continental shelf break covering the entire Ross Sea Embayment (Conway et al., 1999). Climate models identified strong cooling over the Ross sea sector, reflecting the increased elevation and albedo of the extended ice sheet (Buizert et al., 2021). We suggest that this enhanced cooling may have affected the deposition site of Larsen BIA. We note that the past ice sheet thickness at Larsen and the upstream distance travelled since deposition are poorly constrained; both of these should impact the magnitude of the reconstructed temperature change. The

temporal isotope slope ($\alpha_T = 0.58$ ‰ $K^{-1}$) is lower than that for any other ice core site (Buizert et al., 2021) and smaller than the spatial regression slope of around 0.8 ‰ $K^{-1}$ (Masson-Delmotte 2008). However, the temporal isotope slope we reconstructed is likely to be a lower bound because the sublimation might have enriched the $\delta^{18}O_{ice}$ of the glacial (downstream) part of Larsen BIA. The reconstructed surface temperature change of 15 ± 5 °C assumes that the selected tie points are correct. However, if the $\delta^{18}O_{ice}$ features used for the matching are not climatic in origin but rather reflect local effects (such as

sublimation intensity, and accumulation controls by surface slope), then our tie points would be incorrect. Future measurements of ice chemistry and dust loading may improve our Δage estimates, which will allow a refined estimate of past glacial cooling.

From 24.3 ka BP to 10.6 ka BP, the accumulation rate increased by a factor of 1.7–4.6 (from 0.033 ± 0.007 to 0.103 ± 0.042 m ice $yr^{-1}$). The accumulation rate at TALDICE and EDC began to decrease transiently around 14.5 ka BP following the

Antarctic Cold Reversal (ACR), while the reconstructed accumulation rates at the deposition site of Larsen BIA and Taylor

Dome keep increasing across the ACR. We acknowledge that the accumulation rate of the deposition site of Larsen ice younger than 14 ka BP remains poorly constrained because of the large uncertainty, but is highly constrained for the older part (> 14 ka BP). The accumulation rate at the deposition site of Larsen ice is lower than that of TALDICE during 14–21 ka BP and exceeds the accumulation rate of TALDICE after 14 ka BP. The Ross Ice Shelf (RIS) retreated during the last deglaciation (Ship et al., 1999; Yokoyama et al., 2016), and as the RIS retreats, the storm track migrates to the Southern Victoria Land from

the northern part and increases precipitation to the site (Morse et al., 1998; Aarons et al., 2016; Yan et al., 2021). Therefore, we speculate that the storm track affects the original deposition site of the Larsen BIA more than the Talos Dome after 14 ka BP. This interpretation may help studies for reconstructing past atmospheric circulation associated with the retreat of RIS. Likewise, a strong accumulation increase across the last deglaciation was seen at the coastal Law Dome site in the Indian Ocean sector (Van Ommen et al., 2004), also attributed to increases in storm-derived precipitation. However, spatial difference

of the original deposition site of the upstream and downstream Larsen ice is not constrained; this must be known for better interpretation of the accumulation rate.

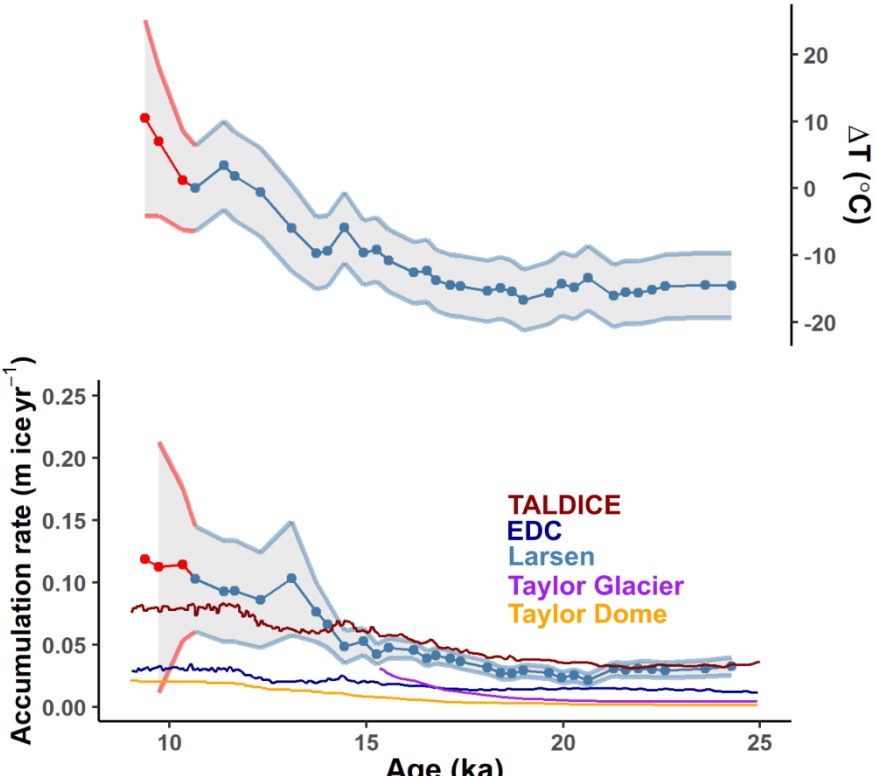

**Figure 10. Reconstructed surface temperature and accumulation rate.** Accumulation rates of TALDICE, EDC are from Veres et al. (2013). Accumulation rates of Taylor Dome, and Taylor Glacier are from Baggenstos et al. (2018). The three youngest values of Larsen blue

ice are rejected (red line) due to the large uncertainty in the Δage that translates into the possibility of negative accumulation rates and an unexpected 10.5 °C increase in ~1200 years. The ΔT is a relative value to 10.6 ka BP. The uncertainty for the ΔT and accumulation rate (1σ) was estimated through the error propagation formula.

# 4 Conclusions

Based on the shapes of dust bands, GPR profile, and analytical data ($\delta^{18}O_{atm}$, $\delta^2H_{ice}$, and $CH_4$) we conclude that the ages of the downstream ice (from ice cores #23 to #200) in Larsen BIA monotonically increase along the ice flow direction. The vertical profile of $\delta^2H_{ice}$ in core #23 correlated well with the record from the horizontal surface ice samples, showing that the stratigraphy of ice is not significantly disturbed and does not cause an age inversion at the site. The negative d-excess values indicate sublimation of the ice and the possibility of enrichment in the $\delta^2H_{ice}$ and $\delta^{18}O_{ice}$ values during sublimation. $CH_4$ is well-preserved in ice core #23 at a depth of > 4.6 m, and gas isotopes ($\delta^{18}O_{atm}$, $\delta^{15}N$-$N_2$) are well-preserved at 1.95 m depth. Horizontal $\delta^{18}O_{atm}$ values at a depth of 1.95 m in the Larsen BIA reveal a typical glacial termination (the Larsen $\delta^{18}O_{atm}$ shows both negative and > 1.0 value). Along with the stable water isotopes, correlation of $CH_4$ concentration, and $\delta^{18}O_{atm}$ with existing ice core records, the gas ages of the studied Larsen ice cover 9.2–23.4 ka BP and ice ages cover 5.6–24.7 ka BP. In addition, the [81]Kr ages from #23 and TF cores also support the ages. We provide high-precision ages for ice and fossil air trapped in blue ice at the Larsen Glacier. Because well-constrained ages and high quality of proxies are essential for paleoclimate studies, our work may help future studies on BIAs across the Antarctic continent. For example, using the ice samples from Larsen Glacier, we suggest a tentative climate reconstruction of surface temperature (15 ± 5 °C increase) and accumulation rate (increased by a factor of 1.7–4.6) at the deposition site of Larsen ice during the last deglaciation. Using the ice samples from Taylor Glacier, Allan Hills, and Larsen Glacier together may also provide information regarding the growth and/or retreat of Ross Ice Shelf. In addition, various sites may be beneficial since the ice quality for paleoclimate studies depends on glaciological conditions. In particular, since a large amount of ice with the same age is exposed at the surface in the BIA, this study may facilitate further research on paleoclimate that was difficult because of the limited amount of ice samples.

**Appendix A: CH₄, CO₂, δ¹⁸Oₐₜₘ, and δ¹⁵N-N₂ measurements of the Larsen Glacier**

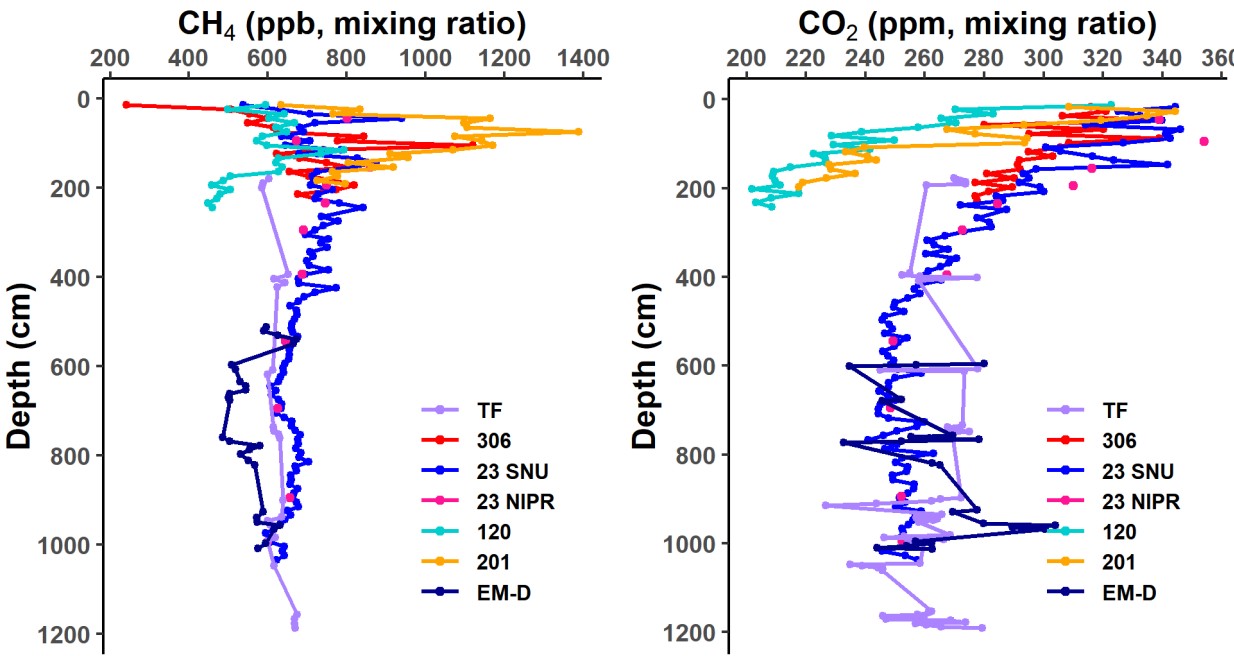

**Figure A1. Vertical profiles of greenhouse gas concentrations at Larsen BIA ice cores.** (a) CH₄ records. (b) CO₂ records. CH₄ was measured using wet extraction, while CO₂ via both dry and wet extraction methods at SNU and NIPR, respectively. Results of TF core are from Jang et al. (2017).

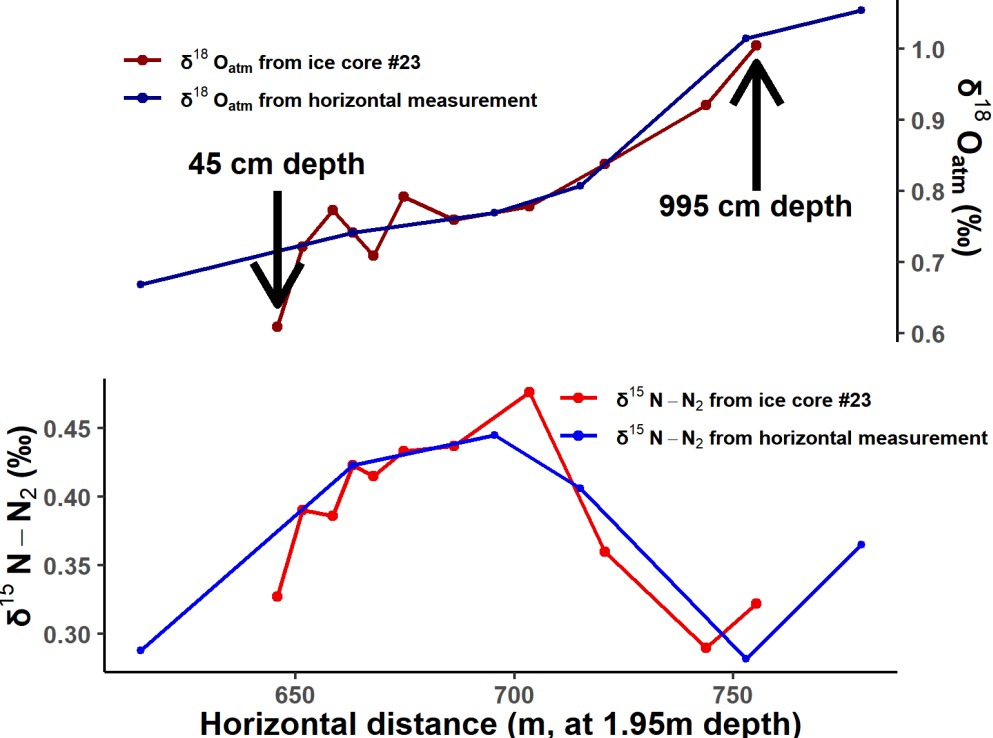

**Figure A2. Nitrogen and oxygen gas isotope record of ice core #23 and horizontal measurement.** Horizontal measurement is conducted using ice cores at 1.95 m depth. Depth of ice core #23 was converted to horizontal distance at 1.95 m depth by using Eq. (3). The uncertainty of measured $\delta^{18}O_{atm}$ at a depth of 1.95 m is assumed to be $\pm$ 0.05 ‰, which was deduced by the offset between the dark-red and dark-blue line.





**Appendix B: Gas age estimation for the Larsen Glacier**

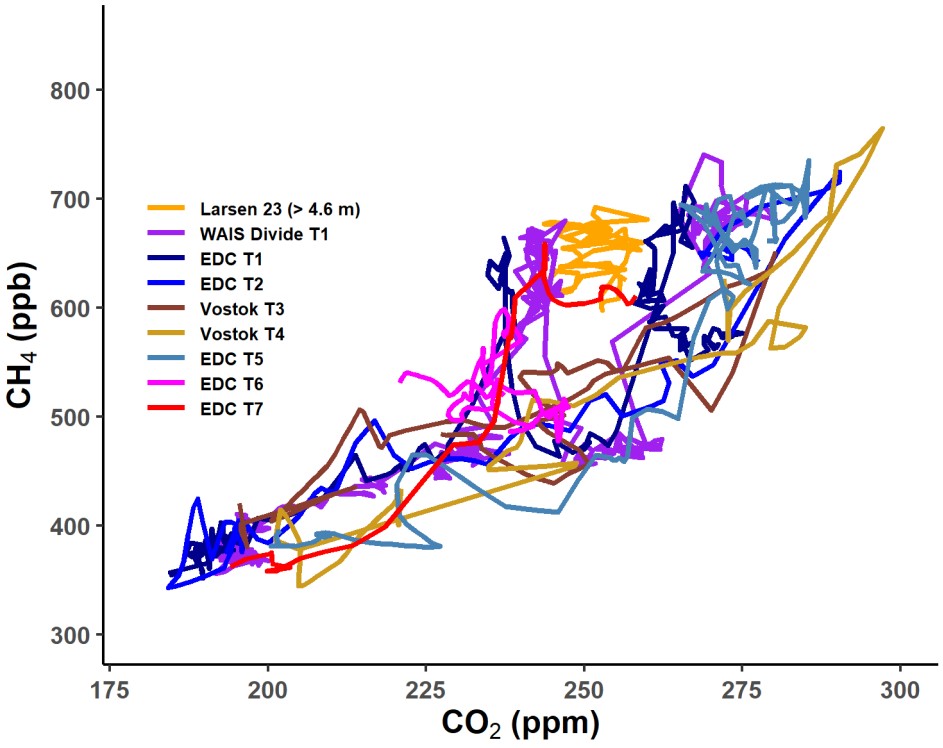

**Figure A3. Comparison of CO₂−CH₄ relationship in the Larsen ice core #23 with existing records during glacial Terminations.** WAIS Divide: Marcott et al. (2014); Rhodes et al. (2017); EDC: Bazin et al. (2013d); Monnin et al. (2001); Monnin et al. (2004); Siegenthaler et al. (2005); Lourantou et al. (2010); Schmitt et al. (2012); Bereiter et al. (2015); Vostok: Petit et al. (1999).

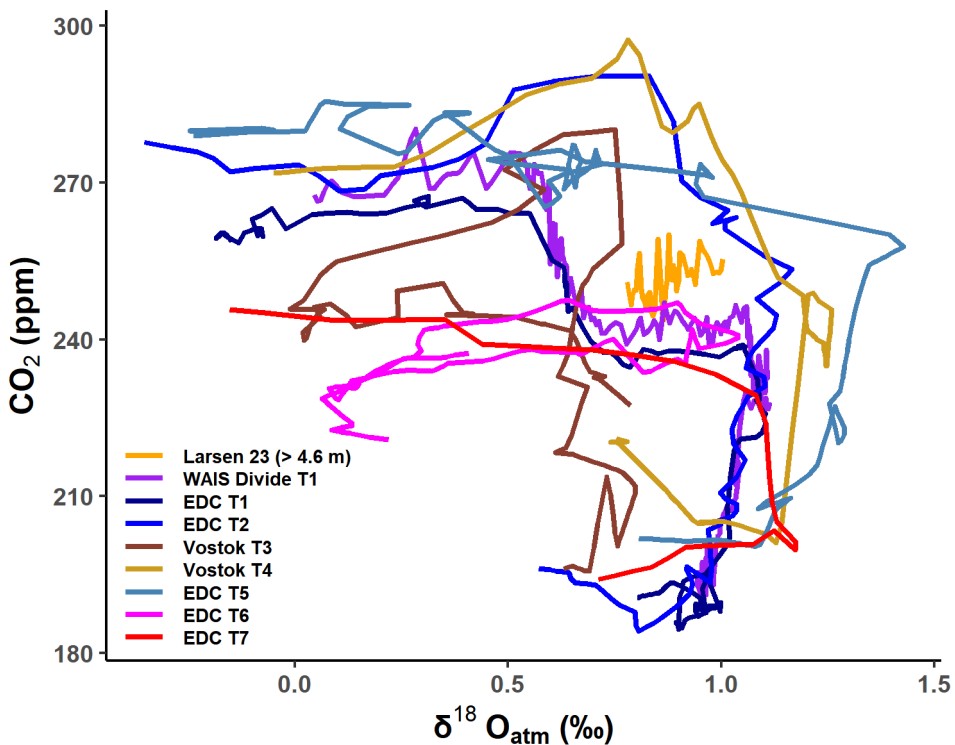

**Figure A4. Comparison of δ¹⁸O_atm–CO₂ relationship in the Larsen ice core #23 with existing records during glacial Terminations.** WAIS Divide: Marcott et al. (2014); Severinghaus et al. (2015); EDC: Monnin et al. (2001); Monnin et al. (2004); Siegenthaler et al. (2005); Lourantou et al. (2010); Schmitt et al. (2012); Landais et al. (2013); Bereiter et al. (2015); Extier et al. (2018a); Vostok: Petit et al. (1999); Bender (2002).

## Appendix C: Ice age estimation for the Larsen Glacier

To estimate the ice age at a depth of 1.95 m, the age/depth relationship of each ice core sample must be determined. The relationship was deduced by a simple calculation using Eq. (A1) and (A2); α, β, γ: ice age at each location; D: horizontal distance between two surface ice samples (at a depth of 20 cm); θ is the average dip of the ice layer; H + 0.2 m: depth where the age is the same as β. The average dip was estimated from the GPR profile (yellow box in Fig. 3c). The average dip of the ice layer located near core #23 was 4.96°, as inferred from Fig. 4b. Then, the ice age at a depth of 1.95 m was calculated using Eq. (A3). Refer to Fig. A5 to understand the following equations:

$H = D \times \tan\theta$ (A1)

Age/depth relation $= (\beta-\alpha)/H$ (A2)

$\gamma = \{(\beta-\alpha)/H\} \times (1.95-0.2) + \alpha$ (A3)

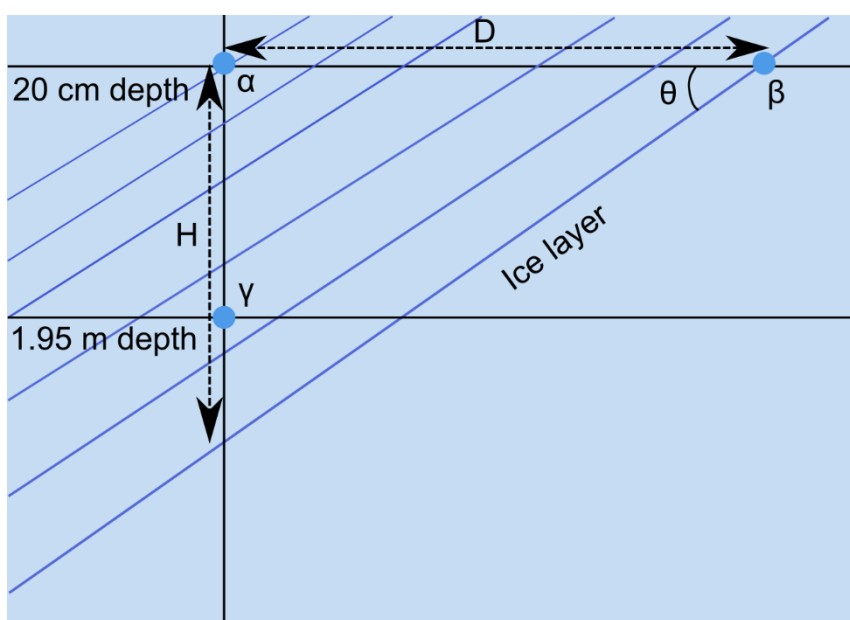

**Figure A5. Schematic illustration for mathematical relation between ice ages.**

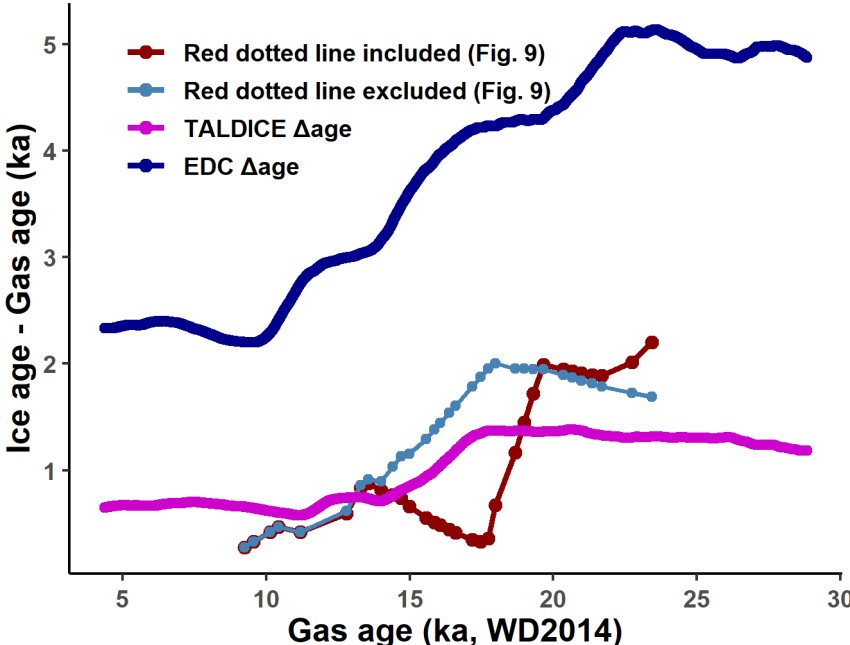

**Figure A6. EDC, TALDICE and Larsen BIA Δage.** Δage for Larsen BIA is defined as (ice age) − (gas age) at 1.95 m depth. EDC WD2014 chronology is from Buizert et al. (2021).

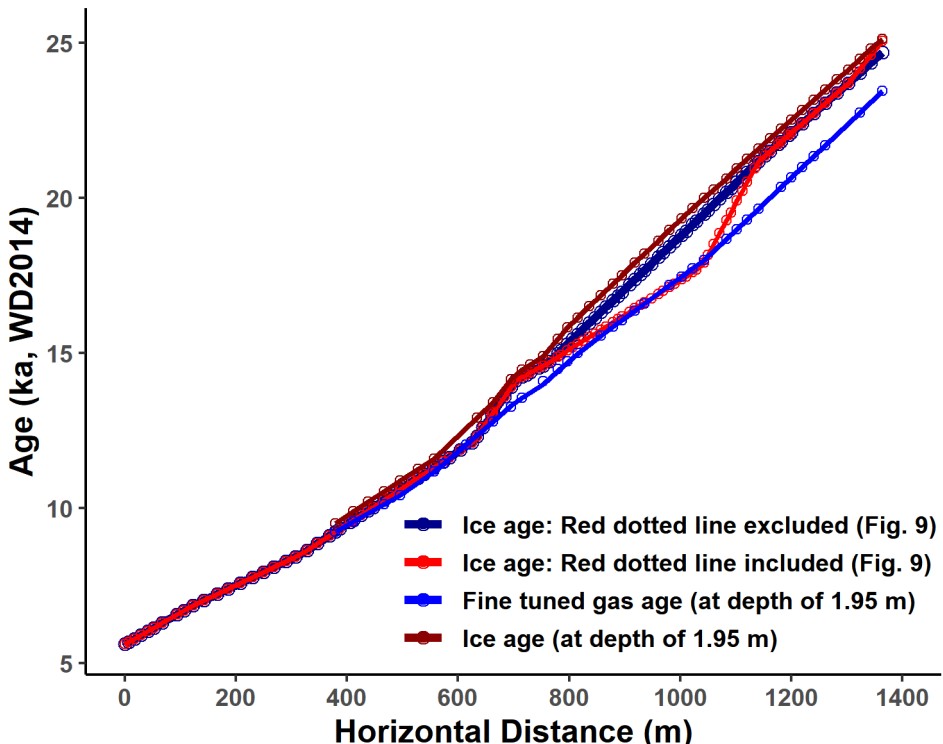

**Figure A7. Relation between horizontal distance and age.**

**Author contributions**

GL and JA conceived the idea of this study, performed experiments, interpreted the data, and wrote the manuscript with contributions from all co-authors. HJ performed ground penetrating radar surveys and data processing. FR, Z.-TL, WJ, and GY carried out [81]Kr age dating of the Larsen Glacier. IO, SG, and KK assisted gas measurements at the NIPR and interpreted the results. SK and JM measured stable water isotopes of the ice. CB established the WD2014 timescale of TALDICE and helped with climate reconstructions and interpretation. SH, CHH, and SDH helped collect ice samples and organize the early plans of this study. All authors have discussed and interpreted the results and contributed to the manuscript.

**Competing interests**

The authors declare that they have no conflict of interest.

**Acknowledgements**

We thank Sang-Young Han, Yoojung Yang, Youngjoon Jang, and Yeongcheol Han for their assistance in collecting ice and
organizing data in the early stages of the study. We sincerely thank Gwangjin Lim, Junghwa Hwang, and Jaeyoung Park for
their laboratory assistance and technical support. We thank Joohan Lee for providing GPR survey equipment. We also thank
the Norwegian Polar Institute for sharing the QGIS Quantarctica package. We also appreciate the Meteo-Climatological
Observatory at MZS and Victoria Land of PNRA for annual surface temperature data at the Larsen Glacier.

**Financial support**

This study has been supported by the Korea Polar Research Institute (grant no. PE21100) and National Research Foundation
of Korea (NRF-2018R1A2B3003256). This work has also received financial support from the National Key Research and
Development Program of China (2016YFA0302200), National Natural Science Foundation of China (41727901), the Japan
Society for the Promotion of Science (JSPS) and the Ministry of Education, Culture, Sports, Science and Technology of Japan
(MEXT) KAKENHI grant numbers 17K12816 to IO; 17H06320 and 15KK0027 to KK, NIPR International Internship Program
for Polar Science to GL, and the U.S. National Science Foundation (ANT-1643394) to CB.

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
