# Peer review of "Chronostratigraphy of Larsen blue ice area in Northern Victoria Land, East Antarctica, and its implications for paleoclimate"

_The Cryosphere, 2021_

## Author Response (AR1)

We thank referee #1 and #2 for helpful comments.

**To referee #1**

Lee et al establishes the chronology of a blue ice field at the Larsen Glacier in north Victoria Land, East Antarctica by cross-correlating properties recorded in the ice (and the enclosed air). Further aided by absolute radiometric $^{81}$Kr dating, the authors report the discovery of a horizontally continuous ice section spanning from the early Holocene through the Last Glacial Maximum, with the age gets progressively older downstream. It is therefore concluded that Larsen Glacier could serve as a paleoclimate archive to study the transition from the Last Glacial Maximum to the Holocene. While the study subject of this manuscript (blue ice) is clearly part of the cryosphere, hence making the manuscript within the scope and aim of the journal The Cryosphere, the manuscript would benefit from more in-depth discussion on the glaciological or climatic implications of the discovery of stratigraphically continuous blue ice at the Larsen Glacier: what does it mean for the local ice dynamics, East Antarctic Ice Sheet, or paleoclimate (given the authors argue this blue ice field could be utilized to study climate changes across the Last Deglaciation)?

➢ In this manuscript, we described the stratigraphy and established the unknown ice and gas ages for the Larsen blue ice. Our chronostratigraphic study for the outcropped ice will serve as a groundwork for future study on Larsen Glacier. Well-constrained chronology for the outcropped ice will enable studies that have been difficult by the limits of availability of large ice samples.

➢ We added a reconstruction of past accumulation rate and surface temperature in section 3.7. In addition, we also added a description of the greenhouse gas alteration at the very shallow depth in BIAs with comparing the gas alteration with other BIAs in section 3.3. For the ice age and gas age uncertainties, we added section 3.6.

It must be acknowledged that a continuous blue ice section is exciting and rewarding for all the field and lab work that was done, but LGM (or Termination I) isn't a particularly understudied interval. A large number of deep ice cores from Greenland and Antarctica have provided a detailed record of atmospheric composition and local to regional climate.

➢ Lots of studies were published in terms of the LGM and the last glacial termination. However, the associated climatic processes remain unclear partly because of the limited size of ice samples, especially for isotopic analysis of greenhouse gas and trace elements. Huge volume of ice sample is allowed from BIAs given that the chronology of the ice is well-constrained.

A blue ice field in Taylor Glacier in the Dry Valleys, less than 500 km away from the Larsen Glacier, provides a near continuous surface ice record already providing large-volume samples for various novel geochemical analyses. Therefore one has to wonder what new information blue ice field in Larsen Glacier could bring about.

➢ Distance between Larsen Glacier and Taylor Glacier is about 330 km and we think this is not a small distance. In addition, when considering the original snow deposition sites, the spatial difference between the Larsen

and Taylor glacier might be larger. Further study in Larsen BIA will give ice flow information of the Northern Victoria Land, while Taylor Glacier will give information about the Southern Victoria Land. Using the ice samples from Taylor Glacier, Allan Hills, and Larsen Glacier together, may also give information about climate conditions in the past as well as the growth and/or retreat of Ross Ice Shelf during the last deglaciation. In addition, we may get benefits from various sites because the ice quality for paleoclimate study depends on glaciological conditions.

➢ We added description of the gas alteration at the very shallow depth in BIAs and compared it with other BIAs (Elephant Moraine, Allan Hills, Taylor Glacier) in section 3.3.

A few questions that may be worthy of consideration: Can you trace the original deposition site by GPR and dust bands? Or if you already know where the ice was deposited, could you estimate the velocity of ice motion? In terms of climate, presumably you could infer annual layer thickness from GPR and that should provide information about past accumulation rates and ice thinning function. If so, what does it mean for the local climate and ice dynamics?

➢ To trace the original deposition site, we should operate an ice flow modeling and conduct further GPR surveys. For study in this aspect, the work in our manuscript will give fundamental information.

➢ Estimation for past accumulation rates and surface temperature can be derived from using $\Delta$age (ice age and gas age difference), and $\delta^{15}$N-$N_2$. We added the estimated values in section 3.7.

Since both hydrogen and oxygen isotopes in water have been measured, could you calculate the deuterium excess and what does that tell us about the hydrological changes in north Victoria Land on glacial-interglacial timescales?

➢ Line 284: We added "Deuterium-excess (d = $\delta^2$H$_{ice}$ – 8 × $\delta^{18}$O$_{ice}$) shows a wide range (5.40 to −3.89 ‰, Table S5) from the entire near-surface ice samples. The negative d-excess likely indicates that isotopic fractionation is attributed to the sublimation of ice in the accumulation zone (Hu et al., 2021). Negative d-excess values were also observed in the Allan Hills BIA (Hu et al., 2021). Meanwhile, sublimation can deplete $^{16}$O and $^1$H in ice and make the isotopic ratios ($\delta^{18}$O$_{ice}$ and $\delta^2$H$_{ice}$) enriched. Thus, the wide range and negative value of d-excess results indicate that stable water isotopes are not proper proxies for the changes in temperature and vapour sources."

This is not to say that these are the only questions that must be answered here. The bottom line is that as a reader of The Cryosphere I am hoping to see what new scientific discovery is being made. It might be an abrupt change in accumulation rates, or a different local precipitation regime. The current manuscript feels to me more like a detailed progress report without firm conclusion on what new is being presented. Of course it could be argued that the discovery of a potentially useful paleoclimate archive itself is an achievement, but back to my earlier points, the Last Glacial Maximum is already an intensively studied interval.

- Taking the advantage of BIA (using huge amount of ice) is essential to discover new insights for the last deglaciation. Therefore, well-constrained chronology for the outcropped ice will enable studies that have been difficult by the limits of availability of large ice samples.
- To find very old ice (> 1 Ma BP), BIAs are the major targets recently. However, well-constrained chronostratigraphic study of BIAs is scarce. Thus, the manuscript here is not just a detailed progress report but a fundamental work and a pioneer of BIA study. The manuscript highly contributes to the ice core society for finding very old ice in BIAs.
- We added a comparison of depths of the un-altered gas between BIAs in section 3.3.
- We added the estimated value of past accumulation rate and surface temperature and the interpretation of it in section 3.7.

Finally, before proceeding to detailed comments, I feel a bit confused why the manuscript does not present the absolute dating results first. $^{81}$Kr is a well-established absolute dating method for glacial ice and underground water. Unless the authors are worried about contamination of modern air (a hypothesis that was later rejected based on undetectable $^{85}$Kr), the results of absolute dating (high accuracy, low precision) should come before the cross-dating efforts that have a high level of precision. In doing so you could easily narrow the range of age search to the last glacial cycle and therefore shorten a considerable portion of the current discussion (in particular 3.5) that might be devoted to more glaciological-focused discussion.

- We used the $^{81}$Kr dating method later in the manuscript, serving as an independent age constraint and strengthening the age constraint established by correlations with existing ice core record. Even though the modern air contamination could be rejected, the use of $^{81}$Kr isotopes for accurate age constraint could be limited because of the age uncertainty originating from inaccurate calibration, estimation of the Kr half-life, and the production rate of Kr through the past.

--

Specific comments:

Line 18: the claim of a "simple stratigraphy of ice" seems to contradict the description that the ice upstream has age repetitions (i.e. is folded). Perhaps you could rephrase it into something like "Here we report a surface transect of ice that has a simple horizontal stratigraphy." This would exclusively correspond to the downstream section described in the current manuscript.

- Line18: We rephrased the sentence.
  - "Here, we report a surface transect of ice with an undisturbed horizontal stratigraphy from the Larsen BIA, Antarctica, making the area valuable for paleoclimate studies."

Line 32: please add Lüthi et al (2008) Nature and Bereiter et al (2015) GRL to the citation.

➢    Line 36: We added "Lüthi et al., 2008; Bereiter et al., 2015".

Line 49: it is necessary to point out that the current longest continuous ice core record stops at 800,000 years.

➢    Line 54: We added "(to date, the longest continuous ice core record covers the last 800 ka BP)".

Line 60: this sentence is equivocal. Does "globally well-mixed" also apply to glaciological records? Based on the nature of stable water isotopes I don't the authors imply that the glaciological records are also globally mixed (in fact, they are not). Please (1) consider splitting the gas age and ice age synchronization methods and (2) point out that the age of the gas is different from the age of the ice at the same depth.

➢    Line 76: We split the words related to dating methods for gas and ice ages. The relevant text is reworded as follows:

      ➢    One effective method for dating the gas age is to correlate globally well-mixed atmospheric gas records (e.g., $CH_4$, $CO_2$, $\delta^{18}O_{atm}$) with existing well-dated ice core records (Spaulding et al., 2013; Baggenstos et al., 2017; Menking et al., 2019; Yan et al., 2021). Other methods include the use of stable Ar isotopes (Higgins et al., 2015; Yau et al., 2015; Yan et al., 2019) or radioactive $^{81}Kr$ (Loosli&Oeschger 1969; Buizert et al., 2014; Tian et al., 2019; Crotti et al., 2021), both of which provide independent and absolute age constraints. However, the use of Ar and Kr isotopes have certain limitation for accurate age constraint. The age uncertainty of Ar dating is ±180 ka or 11 % of the age; the uncertainty is originated from the regression line Bender et al. (2008) used. In addition, the ages can be corrupted by the injection of radiogenic $^{40}Ar$ from the continental crust (Bender et al., 2010). The age uncertainty of $^{81}Kr$ dating ranges between 5–20 % of the age depending on the sample age and sample size (Jiang et al., 2020). It also has a systematic age uncertainty of ~5 % due to the uncertainty in the $^{81}Kr$ half-life. For dating ice ages, glaciochemical records (e.g., nss-$Ca^{2+}$, $\delta^{18}O_{ice}$, $\delta^2H_{ice}$) can be used for correlation with existing well-dated ice core records (Baggenstos et al., 2018; Menking et al., 2019). Notably, the ice is older than the gas at the corresponding depth because the gas is isolated and stops mixing with the atmospheric air when the firn completely transforms into ice (Schwander and Stauffer, 1984).

Line 61 & 81: please add Yan et al (2021) Clim. Past to the citation.

➢    Line 66: We added "Yan et al., 2021".
➢    Line 78: We added "Yan et al., 2021".
➢    Line 91: We added "; Yan et al., 2021".

Line 62: if absolute dating methods are effective, readers without sufficient knowledge on their limits may why bother correlating gas-phase and ice-phase properties? It may be better to introduce

absolute dating methods first, then acknowledge their uncertainties, and finally introduce a more precise way of age synchronization.

➢ Line 78: We removed "effective".

➢ Line 80: We added the limintations of Ar and Kr dating.

    ➢ "However, the use of Ar and Kr isotopes have certain limitation for accurate age constraint. The age uncertainty of Ar dating is ±180 ka or 11 % of the age; the uncertainty is originated from the regression line Bender et al. (2008) used. In addition, the ages can be corrupted by the injection of radiogenic $^{40}$Ar from the continental crust (Bender et al., 2010). The age uncertainty of 81Kr dating ranges between 5–20 % of the age depending on the sample age and sample size (Jiang et al., 2020). It also has a systematic age uncertainty of ~5 % due to the uncertainty in the 81Kr half-life."

Line 85: please specify which "area" you are referring to (north Victoria Land?).

➢ Line 70: We changed "area" to "BIAs in Northern Victoria Land.".

Fig 1: Is there a particular reason for the current orientation of the Antarctic continent?

➢ In Figure 3, north is directed to the top and the directions of ice flow and GPR results are presented in the same way. To keep a consistent orientation in the figures, we flipped the Antarctic continent in Figure 1. We think the opposite orientation in Figure 1 may make the readers confused. However, if the editor and the reviewers strongly suggest to change the orientation, we will do that in the revised manuscript.

➢ Line 104: We added "To keep a consistent orientation with the GPR profile in Fig. 3, we flipped the classical map of Antarctica (East Antarctica to the left-hand side)".

Line 154: could you please evaluate the potential of in situ methane production in ice cores with high dust concentrations (Lee et al 2020 GCA)?

➢ We have not measured concentration of ion species such as $Na^+$ or $Ca^{2+}$ so we are not able to discuss excess $CH_4$ with high dust concentrations in this manuscript.

Line 157: please specify what 2nd gas extraction means. Does it imply the refrozen meltwater is melted once again?

➢ Line 182: We added "(refrozen meltwater was melted and refrozen once again)".

Line 169: please specify the temperature of the water trap.

➢ Line 195: We added "(approximately −80 °C)".

Line 189: what does "unclear ice" mean? It is not a common word to describe ice cores. Please elaborate.

➤ Line 217: We changed "removed some unclear ice surface" to "shaved away some blurry ice surface".

Line 240: could you define the origin to which downstream and upstream are referenced against?

➤ Line 276: We added "(from ice cores #23 to #200)".

➤ Line 278: We added "(from 81w to ice core #23)".

Line 261: the possibility of large variations in temperature and vapor sources is an interesting one. Perhaps you could quickly test them using deuterium excess data.

➤ Line 302: We added "Deuterium-excess ($d = \delta^2H_{ice} - 8 \times \delta^{18}O_{ice}$) shows a wide range (5.40 to −3.89 ‰, Table S5) from the entire near-surface ice samples. The negative d-excess likely indicates that isotopic fractionation is attributed to the sublimation of ice in the accumulation zone (Hu et al., 2021). Negative d-excess values were also observed in the Allan Hills BIA (Hu et al., 2021). Meanwhile, sublimation can deplete $^{16}O$ and $^1H$ in ice and make the isotopic ratios ($\delta^{18}O_{ice}$ and $\delta^2H_{ice}$) enriched. Thus, the wide range and negative value of d-excess results indicate that stable water isotopes are not proper proxies for the changes in temperature and vapour sources.".

Line 289: why aren't $d^{15}N-N_2$ and $d^{18}O-O_2$ expected not to be altered substantially? The intrusion of modern air might not be a problem, but there could be gas loss from the ice and hence fractionation.

➤ Line 346: We changed the sentence.

   ➤ "As $\delta^{15}N-N_2$ and $\delta^{18}O_{atm}$ were measured at very shallow depths (~1.95 m), we compared the horizontal results with the vertical distribution of the ice core #23.".

Line 300: the depth at which $d^{15}N-N_2$ and $d^{18}O-O_2$ no longer vary appears to be different at different sites. In Allan Hills BIA gas composition is stabilized below 7 to 10 m (Spaulding et al 2013, Quaternary Res). Can you comment on this variability?

➤ Line 339: We added "Table 2. Depth of unaltered greenhouse gas composition and mean annual temperature of BIAs.".

**Table 2. Depth of unaltered greenhouse gas compositions and mean annual temperature of BIAs.**

| Site | Depth of unaltered greenhouse gas composition (m) | Mean annual temperature (°C) | Reference for depth of unaltered air | Reference for mean annual temperature |
|---|---|---|---|---|
| Elephant Moraine Texas Bowl | > 10[*] | -30.3 | This study (Fig. A1) | KOPRI AWS (76.27° S, 156.71° E) |
| Allan Hills | > 7–10 | -31[†] | Spaulding et al. (2013) | Delisle and Sievers (1991) |
| Larsen Glacier | > 4.6 | -24.4 | This study (Fig. A1) | Antarctic Meteo-Climatological Observatory |
| Taylor Glacier | > 4 | -18[‡] | Baggenstos et al. (2017) | United States Antarctic Program |

| Pakitsoq | > 0.3 | -5.4[¶] | Petrenko et al. (2006) | Climate-data.org |

[*]Depth of unaltered greenhouse gas compositions from Elephant Moraine Texas Bowl remains uncertain due to the lack of data at depth of > 10 m. Mean annual temperature of Elephant Moraine: provided by KOPRI's automatic weather station (AWS) record of year 2020 and 2021. [†]Allan Hills: not provided by an AWS but by stable water isotopes; [§]deduced by vertical profile of $\delta^{15}$N-N$_2$ and $\delta^{18}$O$_{atm}$ values in Allan Hills. [‡]Taylor Glacier: assumed to be comparable with the mean annual temperature of nearby McMurdo station (~100 km away). [¶]Pakitsoq: assumed to be comparable with the mean annual temperature of the nearby town of Ilulissat (~ 40 km away).

➢ Line 357: We added "In contrast, $\delta^{15}$N-N$_2$ and $\delta^{18}$O$_{atm}$ values in the Allan Hills are stabilized below 7–10 m (Spaulding et al., 2013).".

Line 303: it seems that this section could be simplified given your [81]Kr dating results.

➢ We used the [81]Kr dating method later in the manuscript, serving as an independent age constraint and strengthening the age constraint established by synchronization. Even though the modern air contamination could be rejected, the use of [81]Kr isotopes for accurate age constraint is limited because of the age uncertainty originating from inaccurate calibration, estimation of the Kr half-life, and the production rate of Kr through the past.

Line 374-375: the origin of ice age-gas age difference should be introduced in the earlier section.

➢ We moved the statement to line 86. It was also more elaborated.
  ➢ Notably, the ice is older than the gas at the corresponding depth because the gas is isolated and stops mixing with the atmospheric air when the firn completely transforms into ice (Schwander and Stauffer, 1984).

Line 385: the maximum delta-age at 17.5 ka is another interesting observation that could have important paleoclimate implications (Buizert et al 2021, Science).

➢ We estimated the past accumulation rate and temperature with the delta-age and $\delta^{15}$N-N$_2$ (see the new section 3.7 below).

Line 403 & 414: it would be worthwhile to calculate the temporal resolution of the Larsen BIA samples, especially in the horizontal dimension (easy to do given Fig A7). How does that compare to, for example, the Talos Dome ice core record nearby?

➢ Line 457: We added temporal resolution for the Larsen BIA.
  ➢ The established horizontal ice chronology shows that temporal resolution of 10 yr m$^{-1}$ and 17 yr m$^{-1}$ are available for the Holocene (5.6–12 ka BP) and for the last deglaciation (12–24.6 ka BP), respectively, at the surface ice from Larsen BIA. These estimated resolutions are higher than those of the deep ice core TALDICE covering this time interval (~18 yr m$^{-1}$) (Masson-Delmotte et al., 2011).

Line 405: the word "chemical" usually refers to ions in ice cores.

➤ Line 573: We changed "chemical result" to "analytical data".

Line 406: again please provide a clear reference point against which downstream and upstream are defined.

➤ Line 574: We added "(from ice cores #23 to #200)".

Line 412: can you provide more proof to back the claim of "high-precision ages"? It would be helpful if errors associated with cross-correlating different properties could be presented, like what Menking et al (2019) Clim. Past did.

➤ We added "Section 3.6 Age uncertainty". In this section we discussed the gas age, ice age, and Δage (ice age – gas age) uncertainties. See below.

**To referee #2**

**General comments**

The authors present a thorough analysis of ice samples collected in a blue ice area in Northern Victoria Land, East Antarctica. With complementary methods, both in the field, and in the laboratory, the chronostratigraphy of the ice is analyzed. By comparing the results to existing ice core records, a convincing proof of the estimated age of the ice and the gas entrapped in the ice is provided. Together with radar observations, the analyses from a three-dimensional image of age isochrones, which also allows an estimation of the age of ice near the bedrock. Given the importance of Antarctic blue ice areas in the recent developments in the search for the oldest ice, where 2.7-million-year-old ice has been recovered from a blue ice area in the Transantarctic mountains, the manuscript is very relevant for future paleoclimatic studies, and it can help in defining field work practices of sample collection and motivating site selection for shallow ice cores from blue ice areas.

The authors place their manuscript into context through a short literature review, after which the methods are described with clear subheadings. The results and discussion do justify the conclusions that are drawn, after which the paper is shortly wrapped up in a conclusion.

I do think the implications of the analyses are underexposed and need to be further elaborated without reducing the technical details. These technical details are generally clearly described and seem to ensure reproducible research, although the editor must know that I do not have the required laboratory experience/background to criticize this well. I have noted my suggestions below.

**Specific comments per section**

**Abstract**: clear and concise

Line 1: I do think the first line provides a circular argument

➢ We do not understand the comment.

Line 26: BIAs

➢ Line 31: We changed "BIA" to "BIAs".

**1 Introduction**: In general, the introduction gives all necessary background for reading the paper. However, I think the structure is a bit confusing, with paragraphs that do not follow each other in a logical order. Moreover, the last paragraph that introduces the study is too concise and does not clearly bring forward the goal of the research. Proposed solution: I would suggest to merge the paragraph starting on line 77 to the other paragraph on blue ice areas, starting on line 47.

➢ Line 62: We moved the paragraph right after the paragraph that the reviewer suggested.

Moreover, the approaches in the study (now outlined in the last paragraph), could be merged into the paragraphs starting from line 56 and from line 65 and/or the last paragraph can be more elaborate.

➢ Line 107: We more elaborated the last paragraph as follows:
  ➢ "Our study focuses on the chronostratigraphy of ice in Larsen BIA, Antarctica, which may facilitate future research in this region. We describe the ice flow and structure of an ice body using dust bands and ground penetration radar (GPR) surveys, and then assessed the alterations of the measured stable water isotopes, greenhouse gases ($CH_4$, $CO_2$), and gas isotopes ($\delta^{15}N$-$N_2$, $\delta^{18}O_{atm}$). To constrain the unknown gas and ice ages, $\delta^{18}O_{atm}$, $CH_4$, and $\delta^{18}O_{ice}$ were correlated with existing ice core records. We also independently confirmed the ages using the radiometric $^{81}Kr$ dating method. Finally, using the $\delta^{15}N$-$N_2$, and the $\Delta$age (ice age-gas age difference) results, we present the record of surface temperature and accumulation rate. In contrast to the previous studies, which used the Herron-Longway model (Herron and Langway, 1980; Baggenstos et al., 2018; Menking et al., 2019; Yan et al., 2021), we applied a recently developed analytical framework to estimate past surface temperature and accumulation rates (Buizert, 2021)."

Line 42: I think it is important to mention that the flow is redirected. Normally, the ice flows under gravitational forces towards the margins of the continent. Moreover, it is not the bedrock itself that causes the ice to flow upwards, but it is the bedrock geometry (which in some sense is equal to the mentioned basal topographic obstacles). Also, in many cases these obstacles are exposed above the ice (nunataks).

➢ Line 46: We added that the ice flow is redirected and removed "bedrock" from the sentence.
  ➢ "Once ice is deposited, ice flows to the margin of the ice sheet, where it is exposed on the surface since the basal topographic obstacles cause deep glacial flow upward. Moreover, surface snow is ablated by katabatic winds and/or sublimation (Bintanja, 1999; Sinisalo and Moore, 2010)."

Line 52: instead of blue ice, specify that you mean samples taken from blue ice areas. This remark also applies to the rest of the paper.

➢ Line 17: Changed "blue ice" to "blue ice area."
➢ Line 53: Changed "blue ice" to "BIAs".
➢ Line 58: We changed "blue ice" to "ice samples taken from BIAs (hereafter referred to as blue ice)".
➢ Line 59: Changed "blue ice" to "BIAs".

Figure 1: Specify that orange dots represent "a selection" or "examples", as not all BIA where the chronology has been studied seem to be included (e.g., Zekollari et al. 2019)

• Zekollari, S. Goderis, V. Debaille, M. van Ginneken, J. Gattacceca, A. J. Timothy Jull, J. T. M. Lenaerts, A. Yamaguchi, P. Huybrechts, P. Claeys, Unravelling the high-altitude Nansen blue ice field meteorite trap (East Antarctica) and implications for regional palaeo-conditions. Geochim. Cosmochim. Acta. **248**, 289–310 (2019).

➢ We added Nansen BIA and Taylor Dome (TD) to the figure 1.

[Figure]

**2 Study area and methods**: In general, well-structured and clearly described.

Line 97: (Fig. 2a)

➤ Line 119: We changed "Fig. 2" to "Fig. 2a".

Line 97-99: A low mean annual temperature does not guaranty the absence of melt in a blue ice area. We need to know either the standard deviation of this annual temperature, or a maximum/high percentile of the observations.

➤ Line 120: There was a typo. We changed "-27.2 °C" to "−24.4 ± 11.7 °C".

Line 104: Using Quantarctica needs to be acknowledged by also citing the entire dataset and the corresponding paper

Matsuoka, K., Skoglund, A., & Roth, G. (2018). Quantarctica [Data set]. Norwegian Polar Institute. https://doi.org/10.21334/npolar.2018.8516e961

Matsuoka, A. Skoglund, G. Roth, J. De Pomereu, H. Griffiths, R. Headland, B. Herried, K. Katsumata, A. Le, K. Licht, F. Morgan, P. D. Neff, C. Ritz, M. Scheinert, T. Tamura, A. Van De Putte, M. Van Den Broeke, A. Von Deschwanden, Quantarctica, an integrated mapping environment for

Antarctica, the Southern Ocean, and sub-Antarctic islands. Environ. Model. Softw. 140, 105015 (2021).

➢ Line 126: We added "Matsuoka et al., 2018; Matsuoka et al., 2021".

Line 104-105: it is remarkable that the stratigraphy is disturbed upstream. Why does this not have implications on the stratigraphy downstream? What is the cause of the disturbances? Is there a temporal component to this? These questions should be addressed in the results and discussion section.

➢ Generally, many BIAs have a complicate stratigraphy because of the complicate ice flow and basal topography, as we explained in the introduction section. According to the GPR survey and the dust bands, we confirmed that the stratigraphy of the downstream part is not disturbed. However, it is hard to specify why the upstream part is disturbed but the downstream part is not disturbed. We need ice flow modeling studies and clearer GPR profile.

Line 110-113: can be more concise and clearer, something like: reprojected perpendicular to a line parallel to the ice flow direction.

➢ Line 134: We rephrased the sentence.
  ➢ "An imaginary line parallel to the ice flow direction was used to define the horizontal distance, while a perpendicular line from each sampling location to the line parallel to the ice flow direction was projected to identify the intersection point. Each intersection point was then used to measure the horizontal distance from the most upstream sampling site (Fig. S1)."

Figure 2: mention that dust bands are observed in the field and how they are measured (GPS tracks?)

➢ Line 139: We added "observed in the blue ice field. The line marks are derived from what appeared on Google Earth.".

[Figure]

**Figure 2. Location of the Larsen BIA and sample collection. (a)** Location of Larsen BIA, EM-D core, and Jang Bo Go station. **(b)** Magnified map of Larsen BIA including sample locations. Orange lines represent dust bands observed in the blue ice field. The line marks are derived from what appeared on Google Earth. Blue dots represent locations of surface ice samples. Red diamonds are locations of shallow ice cores. Six representative names are shown, red letters for ~10 m long cores (TF and #23), black letters for 2 m long cores (#306, #120, and #200) and surface ice (81w). Z- and S-folds are recognized by the dust bands at upstream ices. The total transect of the ice sample is approximately 1.4 km. **(c)** Schematized cross-section of the transect. Satellite photo of Antarctica is from the QGIS Quantarctica package (Bindschadler et al., 2008).

Line 135: change "interval" to "spacing"?

➢ Line 159: We changed "interval" to "spacing".

Line 136: specify that these are vertical intervals (also in line 146).

➢ Line 160: We added "vertical".
➢ Line 171: We added "vertical".

Line 154: an average offset should be one number, not a range.

➢ Line 179: Daily average offset is not constant. It changes every day. Hence, we provided the range of the daily average offset. We added "which is in the range of 2–20 ppb".

Line 182: in this section I miss the description of the $\delta Ar/N_2$ analyses that are mentioned in the abstract and published in the supplementary materials.

➢ Line 210: We added ", and $\delta Ar/N_2$".
➢ Line 240: We added "Gravity-corrected $\delta Ar/N_{2,gravcorr}$ ($\delta Ar/N_{2,gravcorr} = \delta O_2/N_2 - 12 \times \delta^{15}N$) is listed in Table S1 and Table S2 but was not used in this study.".

Line 228: please briefly specify here why you use the TALDICE ice core in your research.

➢ Line 261: We added "When constraining the ice age of Larsen ice, the TALDICE record was selected to synchronize Larsen $\delta^{18}O_{ice}$ due to its proximity to the upstream direction of the Larsen BIA.".

**3 Results and discussion**: In general, the emphasis of this section seems to be more on the results than on the discussion. To make the manuscript more accessible for a wide readership and to do justice to the analyses performed by the authors, most paragraphs would need some additional sentences that discuss (the implications) of the data.

Line 243: This line should be at the end of the subsection 3.1, as now first the authors explain the stratigraphic profile, then discuss the basal topography and then return to discussing the stratigraphic profile. Also, in Figure 3b, the ice thickness varies between 200 and 320 meter (not 400). Lastly, it would be nice to have a qualitative statement that the ice thickness decreases along the flow and how this relates to the exposure of glacial ice (as mentioned in the introduction).

➢ Line 281: We moved the sentence to the last part.
➢ Line 282: We changed "400 m" to "380 m".
➢ Line 282: We changed "Fig. 3b" to "Fig. 3a, b".
➢ Line 284: We added a statement about the ice thickness.

➢ "The ice thickness decreases as ice flows with increasing bedrock elevation, which is a favourable condition for the ice to be outcropped."

➢ The modified paragraph is as follows:

  ➢ "In the GPR survey, we identified ice layers (or isochrones) in the transect parallel to the ice flow direction (Fig. 3). The dips of the ice layers range from 1° to 6° with a decreasing trend from the upstream to the downstream direction. The ice layers of the radargram were not clearly visible at a depth of < 10 m because of the direct wave signal. We did not observe any stratigraphic folding structure in the ice layers that made age inversion along the ice flow direction in the mid- to downstream areas (from ice cores #23 to #200). Therefore, we expect monotonic and continuous age changes along the ice-flow direction. However, as shown in the dust bands with S- and Z-folds in the upstream area (Fig. 2b), the upstream stratigraphy might be repeated on a scale of tens of meters (from 81w to ice core #23). In addition, the subsurface ice layer in the upstream area (0–800 m from the most upstream side) was not well recognized from the GPR profile (Fig. 3c). It is possible that the noise caused by crevasses, cavities, or cracks could obscure the signals. In addition, accurate data acquisition might have been hindered by antenna tremors or low battery power at severely cold temperatures. The basal topography is well defined from the GPR data; we observed an ice thickness variation of 200–380 m (Fig. 3a, b). The results of the bedrock elevation and ice thickness (Fig. 3a) were obtained using a kriging method, which is a method of interpolation that provides unbiased prediction at unsampled areas (Oliver and Webster, 1990). The ice thickness decreases as ice flows with increasing bedrock elevation, which is a favourable condition for ice to be outcropped."

Line 245: are these crevasses, cavities, or cracks observed during the measurement campaign?

➢ Cracks were observed at the surface but crevasses or cavities were not. However, we cannot exclude the existence of crevasses or cavities in the subsurface. So, we are suggesting the possibility of those factors.

Figure 3: From Figure S1, it does not appear that the GPR has been performed as a grid of flight lines, are the results presented in panel a obtained by interpolation? Moreover, in the text there is no reference/analysis of the data shown in panel a, so I would suggest to either move the panel to supplementary materials or discuss it in the main text.

➢ Line 282: We added "The results of the bedrock elevation and ice thickness (Fig. 3a) were obtained using a kriging method, which is a method of interpolation that provides unbiased prediction at unsampled areas (Oliver and Webster, 1990).".

Line 277: Reconsider combining section 3.3 and 3.4 and renaming it: "analysis of gas entrapped in the ice".

➢ Line 324: We combined section 3.3 and 3.4 and renamed it as "Analysis of gas entrapped in the ice".

Line 252-271: clear and nice balance between results and discussion of results.

Line 259: Please mention the references to the other published ice core records.

➢ Line 298: We added "(Petit et al., 1999; EPICA Community Members, 2004; Stenni et al., 2011)".

Line 265: why do you conduct a linear interpolation? To have measurements at equal horizontal/vertical spacing?

➢ In order to be most objective, we did a linear interpolation. As our data sets are not a continuous measurement, directly pinpointing the data point is subjective.

Line 279: What do you mean by altered? In Figure A1 only large fluctuations can be observed. Proof for altering comes only when discussing the comparison of the results from NIPR to SNU. This textual discussion would be greatly supported by plotting them in a (separate?) figure.

➢ Line 327: The sentence is modified as follows:
    ➢ "The measurement results of the $CH_4$ and $CO_2$ concentrations for the vertical cores are presented in Fig. A1. The shallow ice cores (#306, #120, #201) show that greenhouse gases are significantly altered for the top 2 m, showing out-of-range values of the range of natural greenhouse gas concentrations during the last 800 ka BP: 340–800 ppb for $CH_4$ and 180–300 ppm for $CO_2$."

Line 295: Refer to Figure 4.

➢ Line 352: We added "(Fig. 4b)".

Line 303: Reconsider combining section 3.5 and 3.6 and adding a little introduction that explains your approach of first identifying the glacial termination, then matching the measured isotope and gas concentration profiles with existing (dated) ice cores, and then confirming your findings with the $^{81}$Kr dating.

➢ Synchronizing Larsen to existing ice core record and identification of the glacial termination is a different subject, so providing a separate section of it will be more proper.

Line 319: … > 1.95 m (Fig. A2); the offset….

➢ Line 377: We moved "(Fig. A2)" to the suggested place.

Line 319: It is not clear why the offset may also come from age difference.

➢ Line 377: We added "because some $\delta^{18}O_{atm}$ record is estimated by linear interpolation due to the lack of $\delta^{18}O_{atm}$ record corresponding to the $CH_4$ of core #23".

Line 321: The statement about that it is altered naturally and/or contaminated is rather speculative. It is also in disagreement with section 3.3. In my opinion, this observation is very interesting and deserves further research (could be mentioned as limitation/recommendation).

➢ Line 380: We deleted "Probably, it is altered naturally and/or contaminated, even at depths of 4.6–10 m (Fig. A3, Fig. A4)".

➢ Line 381: We added "$CO_2$ concentration warrants further investigations.".

Figure 7: Consider omitting T3, T5 and T6, and check the color scheme for color blinds.

➢ We changed the color of EDC T2 to blue from dark green.

➢ We want to show $\delta^{18}O_{atm}$- $CH_4$ trend also indicates T1. Thus, we did not omit T3, T5, and T6.

[Figure]

Line 342: (Fig 8d)

➢ Line 401: We changed "Fig. 8c" to "Fig. 8d".

Line 345-355: Did you consider an automated method such as dynamic time wrapping? Also, it would be nice to discuss already in this paragraph the relation between the corrections made to match the horizontal distance to the age and the observed dip angles (as in line 380-384).

➢ We did not consider an automated method. We will add a new section of age uncertainty (see below).

➢ We added the horizontal distance and age relationship of the fine-tuned gas age and 1.95 m depth ice age in figure A7.

➢ Line 417: We added "The age/distance relationship was reasonable (Fig. A7), showing no abrupt change, as supported by the gradual change in the ice layer dip (Fig. 3c).".

[Figure]

Line 371: I do not understand how biases in the $\delta^{18}O_{ice}$ record are avoided by interpolating the original record.

➢ Line 431: We added "spline curve of the".

Line 374: This statement is not clear and can be elaborated.

➢ We moved the statement to line 86. It is also more elaborated as follows:

➢ "Notably, the ice is older than the gas at the corresponding depth because the gas is isolated and stops mixing with the atmospheric air when the firn completely transforms into ice (Schwander and Stauffer, 1984)."

Figure 9: Panel d does not show much more detail and could be omitted.

➢ We removed panel (d).

[Figure]

➤ Line 467: We removed "(d) $\delta^{18}O_{ice}$ record of TALDICE shown in more detail (Stenni et al., 2010, Bazin et al., 2013c)."

Line: 398-403: This paragraph sounds more like a part of the conclusion.

➤ Line 453: We removed the paragraph.

Line 398: The first sentence undersells the results. It can be a valuable (but obvious) recommendation but needs an explanation of why we would need more precise ages. Moreover, it is not in line with the statement on Line 412.

➤ Line 453: We removed the paragraph.

Line 401: please specify which atmospheric greenhouse gas can be measured at what depth (very relevant for other field work missions).

➢ This is written in line 578–579.

Line 401-403: nice and clear statement.

**4 Conclusion**: The conclusion section can be more elaborate. I suggest including the last paragraph of the previous section (line 398-403). Moreover, an estimation of the horizontal relationship between distance and age (i.e., xxx year/horizontal m), would be informative for other studies at blue ice areas.

➢ We think the paragraph should be in the chronology section.

➢ Line 457: We added temporal resolution for the Larsen BIA.

    ➢ "The established horizontal ice chronology shows that temporal resolutions of 10 yr m$^{-1}$ and 17 yr m$^{-1}$ are available for the Holocene (5.6–12 ka BP) and for the last deglaciation (12–24.6 ka BP), respectively, at the surface ice from Larsen BIA. These estimated resolutions are higher than those of the deep ice core TALDICE covering this time interval (~18 yr m$^{-1}$) (Masson-Delmotte et al., 2011)."

Line 409-410: would be nice to guide the reader along the blue ice area and explain why the observations reveal a very typical glacial termination (as for instance Line 304-305, and the mention of the Antarctic Cold Reversal). Moreover, the Δage along the flowline (Figure A6) can be included in this explanation.

➢ Line 580: We added "(the Larsen $\delta^{18}O_{atm}$ shows both negative and > 1.0 value)".

Line 414: not only on blue ice areas in the Northern Victoria Land. The comprehensiveness makes it a valuable study for BIAs across the Antarctic continent.

➢ Line 585: We changed "blue ice in the Northern Victoria Land." to "on BIAs across the Antarctic continent.".

**Appendices**: clear and concise.

Figure A3, A4: Consider omitting T3, T5 and T6.

We wanted to show that $CO_2$-$CH_4$ and $CO_2$-$\delta^{18}O_{atm}$, also indicate Termination 1. Hence, should be included.

➤ We added new sections.

[revised manuscript text omitted]

---

## Referee Report (RR1)

**General comments**

The new version of the manuscript has been greatly improved, with an elaborate discussion in section 3.7 and a concise conclusion. By adding paragraph 3.7, the implications of the analyses are illustrated and placed into context. I suggest moving the newly added paragraph 3.6 to supplementary materials, given its methodical/technical character. I have noted my specific suggestions per section below.

**Specific comments per section**

Line 1: The word "because" is out of place. I would rephrase as: "In BIAs deep ice outcrops, allowing for the cost-effective collection of large-sized old ice samples at the surface."

Line 46: I do not understand what is meant by "deposited". Also, in the current formulation it seems like ice is exposed on the surface everywhere. Please consider a formulation like "In BIAs, old ice is exposed on the surface. Normally, ice forms through the compaction of snow, and flows directly towards the margin of the ice sheet. However, in some areas, (basal) topographic obstacles redirect the flow of the ice towards the surface, resulting in so-called blue ice areas. In these BIAs, surface snow is ablated by …."

Line 139: Dust lines are very beautifully visible in Google Earth indeed!

Line 255: "due to" → "because of"

Line 274: please add "(subsurface)" before "crevasses, cavities…"

Line 277: ", which is a method of interpolation that provides unbiased prediction at unsampled areas (Oliver and Webster, 1990)." can be removed

Figure 3a: Thank you for clarifying how the data is interpolated. As there is still no reference to or analysis of the data shown in panel a, I would suggest swapping the panel with figure S1 of the supplementary materials (so Fig. S1 becomes Figure 3a, and Figure 3a becomes Figure S1).

Line 304: So, the interpolation is conducted to increase the resolution of the data? (Please specify the motivation in the text)

Line 330: Interesting, and nice overview in Table 2.

Line 368: "which is in line with our observation of ~0.05 ‰ offset between core #23 and horizontal measurements at a depth of > 1.95 m (Fig. A2)." I do not see this offset in Figure 2A. For me it looks like the values of the core #23 fluctuate around the horizontal measurements (with indeed a value of approx. 0.05 ‰, but in my understanding an offset is something constant (so all values would be too low or too high). Possible correction: "which is within the same order of magnitude of the differences observed between core #23 and horizontal measurements at a depth of > 1.95 m (Fig. A2)."

Line 369: I do not understand to what age difference is referred (difference between ..?) and how that explains an offset.

Line 421: By interpolating data no new information is added, so it does not per definition avoid a bias. I'd suggest to remove: "to avoid bias in the spline curve of the …" and just state that the horizontal resolution of the data has been increased through linear interpolation at 5 m intervals.

Line 445: The comparison to the vertical ice core resolution undersells your results: please emphasize that the ice is available in large quantities at the surface (in contrast to TALDICE).

Line 447: In the previous version a different number was mentioned (156 yr/m) – is this correct?

Table 2/Conclusion: "Larsen Glacier" is not the same area as the "Larsen BIA". Please change all occurrences to "Larsen BIA" to avoid ambiguity.

Section 3.6: Although it is always valuable to discuss uncertainties, the added paragraph is mostly describing a rather complicated method, without discussing implications of the estimated uncertainties. I'd suggest moving the paragraph to supplementary materials and indicating the estimated uncertainties in the main text/figures. Moreover, the description of the used method should be clarified/simplified.

Section 3.7: very nice addition to the paper!!

Line 528: "deposition site" means "upstream accumulation area"?

Line 535: Not only the tie points would be incorrect but matching the two records would not be very sensible. However, given the good match this seems not to be the case: maybe reformulate to something like: "Given the reliable results in matching the collected samples to existing ice core data (ref to figure/paragraph), the δ18Oice features used for the matching are likely climatic in origin and are not strongly influenced by local effects (such as sublimation intensity, and accumulation controls by surface slope)."

---

## Editor Decision (ED1)

**Additional comment for section 4.1**

In section 4.1, you elaborately identify the glacial termination to which the ice belongs by combining various techniques. This is a great interdisciplinary approach, but in some sense, it may also be a bit of an 'overkill'. What do I mean by this? From a theoretical ice flow perspective, you would expect the blue ice that is directly located next to the snow (downstream of it) to be very young: i.e. from the present interglacial. To illustrate this, I attached a (rough) sketch of a transect along the flowline of an ice sheet that I once made. It is not a great illustration, but I hope it helps in explaining the concept I am trying to explain here. For another illustration that explains illustrates this concept, you can also refer to Fig.1 in doi.org/10.1126/sciadv.abj8138

In other words: if I would not have seen any of the analyses presented in your manuscript and you would have told me there is a Termination in the BIA and asked me which one it is, I would have answered with relatively high confidence that this must be Termination I. There can be exception to this if there is complex ice flow upstream, with several blue ice areas being crossed etc, but even then, you would rather expect discontinuities in the blue ice area profile further downstream, rather than having 'old ice' (from an interglacial period that is not the last one) just downstream of the snow-blue ice area transition.

At this stage I am not suggesting that section 4.1 should be strongly revised, but I think that **including an explanation that follows what is described above in this section would be an added value**. To be more specific, you could for instance explain in a few sentences that you expect ice to be relatively young from an ice flow perspective (as the blue ice area seems to be located directly downstream of an accumulation area).

Moreover, having an additional more 'glaciological' interpretation to your section would further justify the publication of this study in The Cryosphere (compared to e.g. other more palaeo-focused journals, such as Climate of the Past).

Thanks a lot for taking this comment into account when revising your manuscript!

[revised manuscript text omitted]

---

## Author Response (AR2)

We thank referee #1 and #2 for helpful comments and for reviewing the first revision.

**To referee #1**

In the revised manuscript titled "Chronostratigraphy of Larsen blue ice area in Northern Victoria Land, East Antarctica, and its implications for paleoclimate", the authors made substantial changes and added new materials to the draft. In particular, a section estimating the site temperature and accumulation rate made the paper more interesting and worthy of publication. Most of my concerns and suggestions to the previous version of the manuscript have been properly addressed. Therefore, I am happy to recommend its acceptance by The Cryosphere after some suggested revisions listed below have been resolved. These suggestions mostly strive to improve the clarity and the accessibility of the paper.

Please note that the line numbers are from the finalized manuscript (version 3), not the one with Author Track Change

Line 3: the institution number for Christo Buizert needs to be changed if you move that author forward.

➢ Line 3: Institution number is changed to "5".

Line 17: "chronostratigraphic studies" should not be complicated? Either you mean that the chronostratigraphy is complicated, or the efforts to study the chronostratigraphy is challenging ("owing to fold and fault structures").

➢ Line 19: We changed "complicated" to "challenging".

Line 27: what does the "± 5" mean? Is it 1σ or the 95% Confidence Interval? Please specify.

➢ Line 28: We added "(1σ)".

Also in Line 27: why do you attribute the deglacial warming to the retreat of the AIS? It seems to suggest that AIS retreat is the whole reason for the observed warming. However, according to Section 3.7, the ice sheet retreat and possible elevation changes might only be responsible for the warming in excess of the 7.1 ± 2 deg C (observed at Talos Dome). The current sentence is somewhat misleading and some clarifying statements are needed here.

➢ Line 28: We deleted the description of the reason about temperature and accumulation increase in the abstract. The new relevant text is as follows:
   ➢ A tentative climate reconstruction suggests a large deglacial warming of $15 \pm 5$ °C ($1\sigma$), and an increase in snow accumulation by a factor of 1.7–4.6.

Line 34: "archives" and "records" are synonyms. You could just get rid of "records".

➢ Line 35: We removed "records".

Line 82-83: here I think you made a good case for using Kr-dating as a complementary tool. Perhaps you could say something like "given these uncertainties we opt to use correlation of … with … as the primary dating tool."

➢ Line 92: We added a sentence as reviewer #1 suggested. The relevant text is as follows:
   ➢ Given these uncertainties, here we opt to use correlation of atmospheric gas records with existing well-dated ice core records as the primary dating tool.

Line 89: it may be helpful to highlight that this approach exists before the advent of absolute dating (e.g. Petrenko et al in 2006 vs Bender et al in 2008).

➢ Line 98: We added a statement. The relevant text is as follows:
   ➢ $CH_4$ and $O_2$ are well mixed in the atmosphere (Blunier et al., 2007), and conducting correlations of both $CH_4$ concentration and $\delta^{18}O$ of $O_2$ ($\delta^{18}O_{atm}$) is a well-known strategy for establishing chronologies of blue ice, which have been used even before the advent of absolute dating method (Petrenko et al., 2006; Baggenstos et al., 2017; Yan et al., 2021).

Line 95-99: the description of Kr-81 dating seems out of place. It may be better if you move it to the earlier paragraph (around Line 80). Also, what's the most basic principle of Kr-81 dating?

➢ Line 84: We moved the description of $^{81}Kr$ dating to line 84. We also added the basic principle of $^{81}Kr$ dating. The relevant text is as follows:
   ➢ $^{81}Kr$ is a cosmogenically-produced radioactive isotope with a half-life of $229 \pm 11$ ka, decaying to $^{81}Br$ via electron capture. It is mixed in the atmosphere within 1–2 years and has no significant sources or sinks, which makes $^{81}Kr$ as an ideal tracer (Oeschger, 1987; Zappala et al., 2020). Oeschger (1987) suggested the potential of $^{81}Kr$ for radiometric ice core dating. However, at that time, $10^5$–$10^6$ kg of ice was required. Owing to the development of Atom Trace Trap Analysis (ATTA), the required ice has continued to decrease (Lu et al., 2014; Tian et al., 2019; Jiang et al., 2020; Crotti et al., 2021).

Line 114: can you justify the use of the new analytical framework? The superiority of the new method is not self-evident and needs to be declared so readers will understand why.

➢ Line 119: We added statements. The relevant text is as follows:
  ➢ In contrast to the previous studies, which used the Herron-Longway model (Herron and Langway, 1980; Baggenstos et al., 2018; Menking et al., 2019; Yan et al., 2021), we applied a recently developed analytical framework, which does not require stable water isotope values to estimate past surface temperatures and accumulation rates (Buizert, 2021); stable water isotope values from Larsen BIA seems altered by fractionation during sublimation of ice (see Sect. 3.2).

Line 175-177: if I understand this correctly, did you calculate the average of the daily offsets first, and then subtract the average value from the measurement each day? If it's the former, please specify the average with 1 standard deviation (on top of the range of 2-20 ppb).

➢ Line 183: We calculated the average offset by measuring the standard air that was injected into four flasks containing bubble-free ice samples when doing measurement each day. The average offset varies day by day. So we present the mean of the daily average offsets. We also present the average of the intra-day standard deviation of the $CH_4$ concentration measured from the control group. We deleted the previous statement. The new relevant text is as follows:
  ➢ (average of the daily average offsets: $11.4 \pm 3.2$ ppb ($1\sigma$); average of the intra-day standard deviations of the control group: $3.7 \pm 1.2$ ppb ($1\sigma$))

Line 233: "$\delta O_2/N_2$,gravcorr of around $-30$ ‰ indicates that the ice is poorly preserved" seems to suggest that all your samples have $-30$‰ $O_2/N_2$ ratios. However, this is not the case. In fact, there is only one sample (#301) that is characterized by such a negative value. Maybe you could just describe the $O_2/N_2$ data first, and then say there is one very negative $\delta O_2/N_2$ sample ($<-30$ ‰), which you decided to reject.

➢ Line 241: We replaced the sentence as reviewer #1 suggested.

Line 238: either in this chapter or earlier (around line 80) you need to briefly describe the principle of Kr-81 dating. For example, what is the process producing [81]Kr? What is the decay product? What is the half-life time (given that you mention its uncertainty)?

➢ Line 84: We added the basic principle of [81]Kr dating.

Line 250: this table belongs to "Results and discussion", as you have presented the actual data measured on the ice.

➤ Line 384: We moved the table and the relevant sentence to section 4.1.

Line 305-307: "The average offset of the values of $\delta^2H_{ice}$ between vertical ice core #23 and the horizontal record is 9.8 ‰ and 0.6 ‰ for $\delta^{18}O_{ice}$. We assume that the average offset is the uncertainty (1σ) of stable water isotope values of the Larsen ice." I don't understand what this means. Do you imply an offset in the stable water isotopes between the horizontal samples and the shallow cores? Fig 4b does not show this offset (or that the offset has been corrected).

➤ Line 312: We changed "offset" to "difference". "Offset" was not the right word.
➤ Line 319: We added a magnified figure in Fig. 4 to help readers' understanding.

[Figure]

Line 353: Section 3.4-3.7 could be separated from the Sections before, and be listed as Discussion because you are using the measurement data to make further inferences. Existing Section 3.1-3.3 can be the new Results section. Table 1 can be moved to here (as a Section 3.3.3. Krypton isotopes) as well.

➢ Line 268: We separated the section as reviewer #1 suggested.
➢ Line 384: We moved the table and the relevant sentence to section 4.1.

Line 360: since you have the Kr-81 results you could be more confident here.

> ➤ Line 382: First of all, we wanted to constrain the age without using the [81]Kr result and then finally used the [81]Kr as an independent dating method. The ages of [81]Kr are mentioned in Line 382 (now more confident). The relevant text is as follows:

> > ➤ Finally, we confirmed the ages of the Larsen ice with [81]Kr dating, indicating 9–41 and 14–43 ka for ice from the TF and #23 cores, respectively (Table 2), and concluded that Larsen ice covers the Last Glacial Termination (LGT, T1).

Line 388: gas and ice age could be split into two sections. This way you could combine the ice age of Larsen ice with the Appendix C.

> ➤ Line 403: We separated the section.
> ➤ Line 442: We moved Appendix C into the new section 4.3.

Line 443: an orphan period.

> ➤ We deleted the orphan period.

Line 455: I am not sure about the length requirement but it seems that the discussion on uncertainty can be moved to the appendices (or supplement).

> ➤ We moved the section to the supplement.

Line 519: again please specify the statistical meaning of the ± 5 deg C.

> ➤ Line 494: We added "(1σ)".

Line 541: change "highly constrained" to "well constrained"

> ➤ Line 520: We changed "highly" to "well".

Line 547: this is an interesting idea worth exploring more. Is it possible to have an array of ice cores along the Transantarctic Mountain? The deglacial retreat of the RIS might be reflected in the timing of the accumulation rate increase.

> ➤ We added new statements about the accumulation rate changes at Taylor Dome as well as that at the TALDICE site because they are located near the Transantarctic Mountains.
> ➤ Line 514: We added a new statement. The new relevant text is as follows:

➢ During the same time period, the accumulation rates at Taylor Dome and TALDICE site increased by a factor of ~15.4 and ~2.4, respectively (Veres et al., 2013; Baggenstos et al., 2018). Based on the three records (TALDICE, Larsen blue ice, Taylor Dome), the accumulation rate has increased more at the Southern Victoria Land than the Northern Victoria Land.

➢ Line 526: We added a new statement. The new relevant text is as follows:

➢ The accumulation rate at the Taylor Dome during 14–21 ka BP shows lower value than that from TALDICE and Larsen ice (Fig. 11). However, the accumulation rate increased by a factor of ~5.2 at Taylor Dome, while those at the Larsen and TALDICE increased by a factor of ~2.1, and ~1.8, respectively. Overall, it appears that as RIS retreats, the accumulation rate at the Southern Victoria Land increases more than the Northern Victoria Land. This interpretation is in line with the previous studies (Morse et al., 1998; Aarons et al., 2016; Yan et al., 2021) and may help studies for reconstructing past atmospheric circulation associated with the retreat of RIS.

Line 559: this is a very long paragraph that is hard to follow. Splitting it into two or three shorter paragraphs may help.

➢ Line 541: We split the paragraph into three shorter paragraphs.

Line 601 (Figure A3): Figure A3 and A4 aim to achieve the same goal. You could merge them into a figure with two panels to save space.

➢ Line 572: We merged them into Fig. A3.

Line 611: you could merge Appendix C with the discussion on ice age in the main text (after it has been separated from the gas ice).

➢ Line 442: We moved Appendix C into the new section 4.3.

In supplement:

Line 6: this information is quite important to the main conclusion of the paper. I suggest you move it to the main text as an Appendix. The supplement will be

➢ Line 583: We moved it into Appendix D.

Line 13-14: the word "constant" is broken into two lines.

➢ The document was corrupted. We solved the problem.

Line 16-17: you mean "… depth at all times"?

➢ The document was corrupted. We solved the problem.

Table S1-S7: I commend the authors for attaching the data here. My only suggestion is that those data could be uploaded to a public depository for easier access.

➢ Line 602: We added "Data availability" section.
➢ We corrected the uncertainty values in Table S3 and S11.

**To referee #2**

**General comments**

The new version of the manuscript has been greatly improved, with an elaborate discussion in section 3.7 and a concise conclusion. By adding paragraph 3.7, the implications of the analyses are illustrated and placed into context. I suggest moving the newly added paragraph 3.6 to supplementary materials, given its methodical/technical character. I have noted my specific suggestions per section below.

➢ We moved the section to the Supplement Information.

**Specific comments per section**

Line 1: The word "because" is out of place. I would rephrase as: "In BIAs deep ice outcrops, allowing for the cost-effective collection of large-sized old ice samples at the surface."

➢ Line 16: We rephrased the sentence as reviewer #2 suggested.

Line 46: I do not understand what is meant by "deposited". Also, in the current formulation it seems like ice is exposed on the surface everywhere. Please consider a formulation like "In BIAs, old ice is exposed on the surface. Normally, ice forms through the compaction of snow, and flows directly towards the margin of the ice sheet. However, in some areas, (basal) topographic obstacles redirect

the flow of the ice towards the surface, resulting in so-called blue ice areas. In these BIAs, surface

snow is ablated by …."

➢ Line 47: We rephrased the sentence as reviewer #2 suggested. The new text is as follows:
   ➢ Once snow is deposited on the surface, ice is formed through the compaction of snow, and flows
      directly towards the margin of the ice sheet. However, (basal) topographic obstacles redirect the ice-
      flow towards the surface in some areas, resulting in so-called blue ice areas. In these BIAs, surface
      snow is ablated by katabatic winds and/or sublimation (Bintanja, 1999; Sinisalo and Moore, 2010).

Line 139: Dust lines are very beautifully visible in Google Earth indeed!

➢ Thank you!

Line 255: "due to" → "because of"

➢ Line 259: We changed "due to" to "because of".

Line 274: please add "(subsurface)" before "crevasses, cavities…"

➢ Line 278: We added "(subsurface)".

Line 277: ", which is a method of interpolation that provides unbiased prediction at unsampled areas

(Oliver and Webster, 1990)." can be removed

➢ Line 281: We removed the statement as reviewer #2 suggested.

Figure 3a: Thank you for clarifying how the data is interpolated. As there is still no reference to or

analysis of the data shown in panel a, I would suggest swapping the panel with figure S1 of the

supplementary materials (so Fig. S1 becomes Figure 3a, and Figure 3a becomes Figure S1).

➢ We better show the distribution of ice thickness and bedrock elevation in Larsen BIA together. We briefly
   described the distribution of ice thickness and bedrock elevation in line 279–283.

Line 304: So, the interpolation is conducted to increase the resolution of the data? (Please specify

the motivation in the text)

- ➤ Line 309: We conducted linear interpolation to assign horizontal distances that correspond to the depths of core #23. The relevant text is as follows:
    - ➤ To match the two $\delta^2H_{ice}$ profiles (measurement of ice core #23 and horizontal measurement of near-surface ice), the depth of ice core #23 was converted to the horizontal distances by pinpointing the deepest result of #23 to the result of ice core #104. The horizontal distances for the rest of the depth of core #23 were assigned by conducting linear interpolation.

Line 330: Interesting, and nice overview in Table 2.

- ➤ Thank you for your comment!

Line 368: "which is in line with our observation of ~0.05 ‰ offset between core #23 and horizontal measurements at a depth of > 1.95 m (Fig. A2)." I do not see this offset in Figure 2A. For me it looks like the values of the core #23 fluctuate around the horizontal measurements (with indeed a value of approx. 0.05 ‰, but in my understanding an offset is something constant (so all values would be too low or too high). Possible correction: "which is within the same order of magnitude of the differences observed between core #23 and horizontal measurements at a depth of > 1.95 m (Fig. A2)."

- ➤ Line 376: We corrected the statement as reviewer #2 suggested.

Line 369: I do not understand to what age difference is referred (difference between ..?) and how that explains an offset.

- ➤ Line 377: We deleted "age difference because some". We also rephrased the sentence and changed "offset" to "difference". The new relevant text is as follows:
    - ➤ Due to the lack of a corresponding $\delta^{18}O_{atm}$ record to the $CH_4$ of core #23, the difference may also come from the $\delta^{18}O_{atm}$ records that were estimated by linear interpolation.

Line 421: By interpolating data no new information is added, so it does not per definition avoid a bias. I'd suggest to remove: "to avoid bias in the spline curve of the …" and just state that the horizontal resolution of the data has been increased through linear interpolation at 5 m intervals.

➢ Line 437: We deleted the statement as reviewer #2 suggested and stated that the horizontal resolution of the data has been increased. The new relevant text is as follows:

➢ As described above, the horizontal resolution has been increased by interpolating the original $\delta^{18}O_{ice}$ records at 5 m intervals.

Line 445: The comparison to the vertical ice core resolution undersells your results: please emphasize that the ice is available in large quantities at the surface (in contrast to TALDICE).

➢ In the $1^{st}$ revision, reviewer #1 suggested calculating the temporal resolution in the horizontal dimension and compare it with TALDICE. We think presenting this would be worthwhile.

➢ Line 475: We added "In addition, Larsen BIA, allows for collecting large quantities of ice from the surface".

Line 447: In the previous version a different number was mentioned (156 yr/m) – is this correct?

➢ Yes. Previously we used 156 yr/m. We corrected the number.

Table 2/Conclusion: "Larsen Glacier" is not the same area as the "Larsen BIA". Please change all occurrences to "Larsen BIA" to avoid ambiguity.

➢ We changed "Larsen Glacier" to "Larsen BIA".

Section 3.6: Although it is always valuable to discuss uncertainties, the added paragraph is mostly describing a rather complicated method, without discussing implications of the estimated uncertainties. I'd suggest moving the paragraph to supplementary materials and indicating the estimated uncertainties in the main text/figures. Moreover, the description of the used method should be clarified/simplified.

➢ We moved the section to the supplement.
➢ We clarified the description.
➢ We corrected the uncertainty values in Table S3 and S11.

Section 3.7: very nice addition to the paper!!

➢ Thank you for your comment!

Line 528: "deposition site" means "upstream accumulation area"?

➢ Line 502: Yes. We rewrote the statement for the readers' understanding. The relevant text is as follows:
  ➢ We suggest that this enhanced cooling may have affected the original deposition site (upstream accumulation area) of Larsen BIA.

Line 535: Not only the tie points would be incorrect but matching the two records would not be very sensible. However, given the good match this seems not to be the case: maybe reformulate to something like: "Given the reliable results in matching the collected samples to existing ice core data (ref to figure/paragraph), the $\delta^{18}O_{ice}$ features used for the matching are likely climatic in origin and are not strongly influenced by local effects (such as sublimation intensity, and accumulation controls by surface slope)."

➢ We agree that the ice age is not significantly incorrect. However, because matching the ice age was relatively difficult with only using $\delta^{18}O_{ice}$, ice age must be improved by matching ion concentrations. Also, it is likely that $\delta^{18}O_{ice}$ value was altered by fractionation during sublimation because the d-excess value indicate negative value (please see section 3.2 and also Hu et al., 2021). Hence, we are cautious to say directly that $\delta^{18}O_{ice}$ value is climatic in origin and that the ice age is reliable.

---

## Author Response (AR3)

Dear Editor,

We sincerely thank for the helpful comments and for reviewing the manuscript. The comments were very valuable and helpful for improving the manuscript and for making it clearer.

The points not written here have been corrected as suggested.

**Editor Comments**

Additional comment for section 4.1

In section 4.1, you elaborately identify the glacial termination to which the ice belongs by combining various techniques. This is a great interdisciplinary approach, but in some sense, it may also be a bit of an 'overkill'. What do I mean by this? From a theoretical ice flow perspective, you would expect the blue ice that is directly located next to the snow (downstream of it) to be very young: i.e. from the present interglacial. To illustrate this, I attached a (rough) sketch of a transect along the flowline of an ice sheet that I once made. It is not a great illustration, but I hope it helps in explaining the concept I am trying to explain here. For another illustration that explains illustrates this concept, you can also refer to Fig.1 in doi.org/10.1126/sciadv.abj8138

In other words: if I would not have seen any of the analyses presented in your manuscript and you would have told me there is a Termination in the BIA and asked me which one it is, I would have answered with relatively high confidence that this must be Termination I. There can be exception to this if there is complex ice flow upstream, with several blue ice areas being crossed etc, but even then, you would rather expect discontinuities in the blue ice area profile further downstream, rather than having 'old ice' (from an interglacial period that is not the last one) just downstream of the snow-blue ice area transition.

At this stage I am not suggesting that section 4.1 should be strongly revised, but I think that including an explanation that follows what is described above in this section would be an added value. To be more specific, you could for instance explain in a few sentences that you expect ice to be relatively young from an ice flow perspective (as the blue ice area seems to be located directly downstream of an accumulation area). Moreover, having an additional more 'glaciological' interpretation to your section would further justify the publication of this study in The Cryosphere (compared to e.g. other more palaeo-focused journals, such as Climate of the Past).

Thanks a lot for taking this comment into account when revising your manuscript!

➤ Line 384: We included an explanation that we expect relatively young ices from Larsen BIA. The new relevant text is as follows:

  ➤ Based on satellite image of the studied area, Larsen BIA might have been located directly downstream of an accumulation zone and the ice might have been originated solely from an accumulation zone because no other BIAs are identified upstream (refer to Fig. 1 in Zekollari et al., 2019). This expectation is supported by the GPR survey profile (Fig. 3b, c) and the $\delta^{18}O_{atm}$ (Fig. 4a), which indicates no complex ice stratigraphy and no discontinuities in the downstream area of Larsen BIA, respectively. In terms of this ice flow condition, we expect the age of Larsen BIA corresponds to relatively young one among the latest glacial terminations.

Line 24: Sampled from the surface (horizontal ice core)

➢ Line 25: Discrete ice samples were collected. Not a horizontal ice core.

Line 26: climatic interpretation is complicated by the need for upstream flow corrections, evidence for strong surface sublimation during the last ice age, and potential errors in the estimated gas age-ice age difference.

➢ Line 28: This sentence is moved to "Line 31".

Line 77: Depends on age of the ice, right? If so, would probably omit.

➢ Line 88: According to Bender et al. (2008), the age uncertainty would be ±180 ka or 11% of the age, whichever is greater. So we think it is necessary to include this in the text.

Line 88: For dating ice ages, glaciochemical records…

➢ Line 100: The sentence is moved to "Line 113".

Line 158: Suggest including this information in section 2.1, rather than here

➢ Line 178: The sentence is moved to "Line 150".

Line 281: and >2200 m?

➢ Line 302: Ice layers at a distance of > 2200 m are identified at Fig. 3c. The new relevant text is as follows:
  ➢ Ice layers at a distance of > 2,500 m have a relatively flat dip.

Line 377: Unclear and does not seem to be consistent. What are the upper limits? What about the lower limits? The errors for the 81Kr age are not only due to atom counting, right?

➢ Line 412: We added which column we were referring to.

Line 387: These are not results from your study here. You use them to compare. Probably better to have these in suppl. mat. Could potentially even consider having some of the figures you have now in suppl mat, but which are quite crucial to your story, in the main text (such as A1)

➢ Line 423: We moved the figure to "Appendix C".